# Stable flow-induced expression of KLK10 inhibits endothelial inflammation and atherosclerosis

**Darian Williams[1,2†], Marwa Mahmoud[1†], Renfa Liu[1,3†], Aitor Andueza[1], Sandeep Kumar[1], Dong-Won Kang[1], Jiahui Zhang[1], Ian Tamargo[2], Nicolas Villa-Roel[1], Kyung-In Baek[1], Hwakyoung Lee[4], Yongjin An[4], Leran Zhang[5], Edward W Tate[5], Pritha Bagchi[6], Jan Pohl[7], Laurent O Mosnier[8], Eleftherios P Diamandis[9], Koichiro Mihara[10], Morley D Hollenberg[10], Zhifei Dai[3], Hanjoong Jo[1,2,11]\***

[1]Coulter Department of Biomedical Engineering, Emory University and Georgia Institute of Technology, Atlanta, United States; [2]Molecular and Systems Pharmacology Program, Emory University, Atlanta, United States; [3]Department of Biomedical Engineering, Peking University, Beijing, China; [4]Celltrion, Incheon, Republic of Korea; [5]Department of Chemistry, Imperial College London, London, United Kingdom; [6]Emory Integrated Proteomics Core, Emory University, Atlanta, United States; [7]Biotechnology Core Facility Branch, Centers for Disease Control and Prevention, Atlanta, United States; [8]Department of Molecular Medicine, Scripps Research Institute, San Diego, United States; [9]Department of Pathology and Laboratory Medicine, Mount Sinai Hospital, Toronto, Canada; [10]Department of Physiology and Pharmacology, University of Calgary, Calgary, Canada; [11]Department of Medicine, Emory University, Atlanta, United States

**\*For correspondence:**
hjo@emory.edu

†These authors contributed equally to this work

**Abstract** Atherosclerosis preferentially occurs in arterial regions exposed to disturbed blood flow (*d-flow*), while regions exposed to stable flow (*s-flow*) are protected. The proatherogenic and atheroprotective effects of *d-flow* and *s-flow* are mediated in part by the global changes in endothelial cell (EC) gene expression, which regulates endothelial dysfunction, inflammation, and atherosclerosis. Previously, we identified kallikrein-related peptidase 10 (*Klk10*, a secreted serine protease) as a flow-sensitive gene in mouse arterial ECs, but its role in endothelial biology and atherosclerosis was unknown. Here, we show that KLK10 is upregulated under *s-flow* conditions and downregulated under *d-flow* conditions using in vivo mouse models and in vitro studies with cultured ECs. Single-cell RNA sequencing (scRNAseq) and scATAC sequencing (scATACseq) study using the partial carotid ligation mouse model showed flow-regulated *Klk10* expression at the epigenomic and transcription levels. Functionally, KLK10 protected against *d-flow*-induced permeability dysfunction and inflammation in human artery ECs, as determined by NFκB activation, expression of vascular cell adhesion molecule 1 and intracellular adhesion molecule 1, and monocyte adhesion. Furthermore, treatment of mice in vivo with rKLK10 decreased arterial endothelial inflammation in *d-flow* regions. Additionally, rKLK10 injection or ultrasound-mediated transfection of *Klk10*-expressing plasmids inhibited atherosclerosis in *Apoe*[−/−] mice. Moreover, KLK10 expression was significantly reduced in human coronary arteries with advanced atherosclerotic plaques compared to those with less severe plaques. KLK10 is a flow-sensitive endothelial protein that serves as an anti-inflammatory, barrier-protective, and anti-atherogenic factor.

## Editor's evaluation

This group has previously demonstrated that endothelial expression of kallikrein related-peptidase 10 (KLK10) is elevated arteries in conditions of high stable flow and down-regulated by disturbed flow conditions. In the present study, the authors tested the anti-atherogenic effects of KLK10 and found that endothelial expression of KLK10 after artery ligation or exposure to oscillatory flow decreases. Notably, recombinant KLK10 or KLK10 overexpression reproduce many vaso-protective effects and reduced atherosclerosis in a mouse model. Besides new insights into the atherogenic process, a potential therapeutic target may have been discovered.

## Introduction

Atherosclerosis is an inflammatory disease that preferentially occurs in branched or curved arterial regions exposed to disturbed flow (*d-flow),* while areas of stable flow (*s-flow*) are protected from atherosclerosis (*Chiu and Chien, 2011*; *Davies, 1995*; *Kwak et al., 2014*; *Tarbell et al., 2014*). Endothelial cells (ECs) are equipped with several mechanosensors located at the luminal and abluminal surface, cell–cell junction, and cytoskeleton, which detect fluid shear stress and trigger cascades of signaling pathways and cellular responses (*Kwak et al., 2014*; *Tarbell et al., 2014*; *Mack et al., 2017*; *Tzima et al., 2005*; *Li et al., 2015*; *Chachisvilis et al., 2006*; *Florian et al., 2003*; *Wang et al., 2016*). *D-flow* induces endothelial dysfunction and atherosclerosis in large part by regulating flow-sensitive coding and noncoding genes, as well as epigenetic modifiers (*Davies, 1995*; *Kumar et al., 2014*; *Kumar et al., 2019*; *Dunn et al., 2014*). Using the partial carotid ligation (PCL) mouse model of atherosclerosis and transcriptomic studies, we identified hundreds of flow-sensitive genes in ECs that change by *d-flow* in the left carotid artery (LCA) compared to the *s-flow* in the right carotid artery (RCA) (*Nam et al., 2009*; *Ni et al., 2010*). Among the flow-sensitive genes, kallikrein-related peptidase 10 (*Klk10*) was identified as one of the most flow sensitive; with high expression under *s-flow* and low expression under *d-flow* conditions (*Ni et al., 2010*). However, its role in endothelial function and atherosclerosis was not known.

*KLK10* was initially identified as a normal epithelial cell-specific 1 (*NES1*) (*Diamandis et al., 2000*) and is a member of the kallikrein-related peptidase 'KLK' family of 15 secreted serine proteases, which are found as a gene cluster on human chromosome (19q13.4) (*Yousef et al., 1999*). The tissue KLKs are distinct from plasma kallikrein, which is encoded on a separate chromosome (4q35) (*Yousef and Diamandis, 2003*). Despite the chromosomal clustering of the KLKs, each enzyme has a unique tissue expression pattern with different cellular functions. Typically, the KLKs are produced as inactive full-length prepropeptides, which are then secreted and activated by a complex process to yield active extracellular enzyme (*Yousef and Diamandis, 2003*). KLKs are involved in a wide variety of processes ranging from skin desquamation to tooth development, hypertension, and cancer (*Madeddu et al., 2007*; *Clements et al., 2004*; *Yousef and Diamandis, 2001*; *Pampalakis and Sotiropoulou, 2007*; *Margolius, 1998*; *Campbell, 2001*).

KLK10 was initially discovered as a potential tumor suppressor with its expression downregulated in breast, prostate, testicular, and lung cancer (*Goyal et al., 1998*; *Liu et al., 1996*; *Hu et al., 2015*; *Luo et al., 2001*; *Zhang et al., 2010*). Further studies, however, showed a more complex story as KLK10 is overexpressed in ovarian, pancreatic, and uterine cancer (*Luo et al., 2003*; *Yousef et al., 2005*; *Dorn et al., 2013*; *Tailor et al., 2018*; *Sotiropoulou et al., 2009*; *Bharaj et al., 2002*). However, the role of KLK10 for endothelial function and atherosclerosis is not known.

Here, we tested the hypothesis that KLK10 mediates the anti-atherogenic effects of *s-flow*, while the loss of KLK10 under *d-flow* conditions leads to proatherogenic effects.

## Results

### KLK10 expression is increased by *s-flow* and decreased by *d-flow* in ECs in vitro and in vivo

We first validated our previous mouse gene array data at the mRNA and protein levels by additional quantitative real-time polymerase chain reaction (qPCR), immunostaining, western blots, and ELISA in ECs in vivo and in vitro. To validate the flow-dependent regulation of KLK10 expression in vivo, mouse

PCL surgery was performed to induce *d-flow* in the LCA while maintaining *s-flow* in RCA (*Figure 1a*). Consistent with our previous data (*Nam et al., 2009*; *Ni et al., 2010*), KLK10 protein (*Figure 1b, c*) and mRNA expression (*Figure 1d*) were significantly higher in ECs in the *s-flow* RCA compared to the *d-flow* LCA. Interestingly, KLK10 protein was also found in the adventitia and occasionally observed in the subendothelial layer as well (*Figure 1b*). In addition, KLK10 protein expression was reduced in the lesser curvature (LC; the atheroprone aortic arch region that is naturally and chronically exposed to *d-flow*) compared to the greater curvature region (GC; the atheroprotected aortic arch region that is naturally and chronically exposed to *s-flow*) as shown by *en face* immunostaining (*Figure 1e, f*).

We next tested whether flow can regulate KLK10 expression in vitro using human aortic ECs (HAECs) exposed to unidirectional laminar shear (LS at 15 dynes/cm$^2$) or oscillatory shear (OS at ±5 dynes/cm$^2$ at 1 Hz) for 24 hr using the cone-and-plate viscometer, mimicking *s-flow* and *d-flow* conditions in vivo, respectively (*Jo et al., 2006*; *Chang et al., 2007*). *KLK10* mRNA (*Figure 1g*), KLK10 protein in cell lysates (*Figure 1h, j*), and secreted protein in the conditioned media (*Figure 1j*) were decreased by OS and increased by LS, confirming the role of KLK10 as a flow-sensitive gene and protein in vivo and in vitro.

We further confirmed the flow-dependent expression of *Klk10* by reanalyzing the single-cell RNA sequencing (scRNAseq) and scATACseq datasets that we recently published using the PCL model (*Andueza et al., 2020*). For the scRNAseq and scATACseq study, single cells and nuclei obtained from the LCAs and RCAs, respectively, at 2 days or 2 weeks after the PCL were used. As described previously, the carotid artery wall cells were identified as EC clusters (E1–E8), smooth muscle cells (SMCs), fibroblasts (Fibro), monocytes/macrophages (Mo1–4), dendritic cells (DCs), and T cells (*Andueza et al., 2020*, *Figure 1k*). E1–E4 clusters consisted of ECs exposed to acute and chronic *s-flow* conditions (2 days and 2 weeks). E5–E7 clusters consisted of ECs exposed to acute *d-flow* (2 days). E8 cells were exclusively found in the chronic *d-flow* condition (2 weeks). As shown in the scRNAseq data analysis (*Figure 1k*; *Figure 1—figure supplement 1*), *Klk10* transcript expression is highest in *s-flow* (E2 and E3) and decreases in response to acute (E5 and E7) and chronic (E6 and E8) *d-flow*. It also shows *Klk10* expression is specific to ECs and not expressed in other cell types studied in the carotid artery. Similarly, scATACseq data (*Figure 1i*; *Figure 1—figure supplement 2*) showed that the *Klk10* promoter region is open and accessible (indicating active transcription status) only in ECs exposed to *s-flow* conditions but closed and inaccessible (indicating inactive transcription status) in ECs under *d-flow* conditions and all other non-EC types. Together, both the scRNAseq and scATACseq results demonstrate that *Klk10* expression is potently regulated by flow in ECs at the epigenomic and transcriptome level, supporting the in vitro and in vivo results shown above (*Figure 1b–g*). Importantly, all non-EC types in the carotid artery express nearly undetectable levels of *Klk10* mRNA transcript and also display closed chromatin accessibility in the *Klk10* promoter region, demonstrating that *Klk10* is primarily expressed by ECs. This suggests that KLK10 protein observed in nonendothelial layers, including the adventitia and subendothelial layer (*Figure 1b*), is unlikely to be originated from cell types other than ECs.

## KLK10 inhibits endothelial inflammation and protects permeability barrier

We next tested if KLK10 regulates EC function by evaluating its role in endothelial inflammatory response, tube formation, migration, proliferation, and apoptosis, which play critical roles in the pathogenesis of atherosclerosis. Treatment of human umbilical vein ECs (HUVECs) with rKLK10 significantly inhibited migration and tube formation, but not proliferation and apoptosis (*Figure 2—figure supplement 1*). In addition, transfection of HAECs with plasmids to overexpress KLK10 reduced THP-1 monocyte adhesion to the ECs in response to tumor necrosis factor alpha (TNFα) and under basal conditions (*Figure 2a*; *Figure 2—figure supplement 2*). Next, we pretreated HAECs overnight with increasing concentrations of rKLK10, followed by TNFα treatment (5 ng/ml for 4 hr). Treatment with rKLK10 significantly inhibited monocyte adhesion to ECs in a concentration-dependent manner (*Figure 2b*). Of note, the anti-inflammatory effect of rKLK10 was lost if rKLK10 was heated, implicating the importance of the enzymatic activity or native conformation of KLK10 (*Figure 2b*; *Figure 2—figure supplement 3*). Furthermore, treatment with rKLK10 significantly inhibited mRNA and protein expression of the proinflammatory adhesion molecules vascular cell adhesion molecule 1 (VCAM1) and intracellular adhesion molecule 1 (ICAM1) (*Figure 2c–g*).

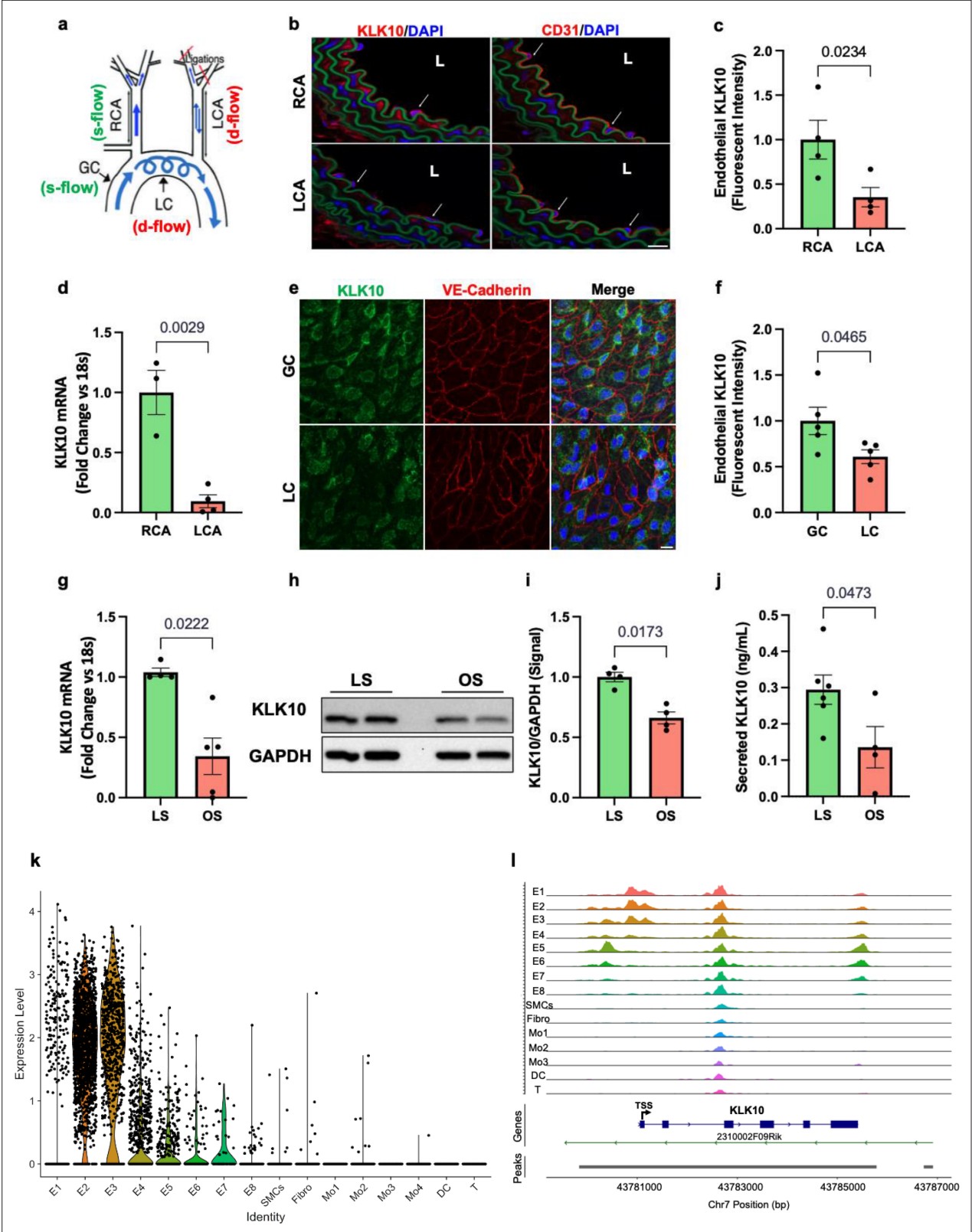

**Figure 1.** KLK10 expression is suppressed by disturbed flow (*d-flow*) and elevated by stable flow (*s-flow*) in endothelial cells (ECs) in vitro and in vivo. (**a**) Depiction of the partial carotid ligation (PCL) surgery and flow-sensitive regions in the aortic arch: right carotid artery (RCA; *s-flow*), left carotid artery (LCA; *d-flow*), greater curvature (GC; *s-flow*), and lesser curvature (LC; *d-flow*). Two days following the PCL of C57BL/6J mice, the RCA and LCA were collected for frozen section imaging (**b, c**) and (**d**) endothelial-enriched RNA preparation. (**b**) Confocal images of immunostaining with anti-KLK10

*Figure 1 continued on next page*

*Figure 1 continued*

or anti-CD31 antibodies (red) and counterstained with 4',6-diamidino-2-phenylindole (DAPI, blue) are shown. Scale bar = 20 μm. Arrows indicate endothelial cells and L is the lumen. (**c**) Quantification of endothelial KLK10 fluorescence intensity expressed as fold-change normalized to the RCA. *N* = 4. (**d**) *Klk10* mRNA was measured in endothelial-enriched RNA from the carotid arteries by quantitative real-time polymerase chain reaction (qPCR). Data are expressed as fold-change normalized to 18s internal control. *N* = 3–4. (**e**) Confocal images of *en face* coimmunostaining of the LC and GC with anti-KLK10 (green) and anti-VE-Cadherin (red) antibody are shown counterstained with DAPI (blue). Scale bar = 10 μm. (**f**) Quantification of endothelial KLK10 fluorescence intensity expressed as fold-change normalized to the GC. *N* = 5. (**g–j**) Human artery endothelial cells (HAECs) subjected to 24 hr of unidirectional laminar shear (LS; 15 dynes/cm$^2$) or oscillatory shear (OS; ± 5 dynes/cm$^2$) were used to measure expression of *KLK10* mRNA by qPCR (**g**), KLK10 protein in cell lysates by western blot (**h, i**), and KLK10 protein secreted to the conditioned media by ELISA (**j**). *N* = 4–6. All data are represented as mean ± standard error of mean (SEM). Statistical analyses were performed using paired *t*-test (***Figure 1—source data 1***). (**k**) Single-cell RNAseq analysis of *Klk10* gene transcripts and (**l**) single-cell ATACseq analysis of *Klk10* chromatin accessibility in eight endothelial cell clusters (E1–E8), smooth muscle cells (SMCs), fibroblasts (Fibro), 4 monocytes/macrophages clusters (Mo1–4), dendritic cells (DCs), and T cells (T) in the mouse carotid arteries following 2 days or 2 weeks of the PCL surgery as we recently reported (***Andueza et al., 2020***). The published datasets (***Andueza et al., 2020***) were reanalyzed here for the *Klk10* gene. E1–E4 clusters represent ECs exposed to *s-flow* conditions in the RCA. E5 and E7 clusters represent ECs exposed to acute (2 days) *d-flow* in the LCA. E6 and E8 clusters represent ECs exposed to chronic (2 weeks) *d-flow* in the LCA. TSS indicates transcription start site.

The online version of this article includes the following source data and figure supplement(s) for figure 1:

**Source data 1.** Western blots for KLK10 and GAPDH.

**Figure supplement 1.** Single-cell RNA sequencing (scRNAseq) analysis of *Klk10* and *Pecam1* from the partial carotid ligation (PCL) mouse model.

**Figure supplement 2.** scATAC sequencing (scATACseq) analysis of *Klk10* and *Pecam1* from the partial carotid ligation (PCL) mouse model.

We then tested the effect of KLK10 on the endothelial inflammatory response under flow conditions in vitro and in vivo. rKLK10 treatment inhibited OS-induced monocyte adhesion in HAECs (***Figure 2h***). In contrast, siRNA-mediated knockdown of *KLK10* significantly increased monocyte adhesion under LS conditions (***Figure 2i***). We next tested if rKLK10 could also inhibit the endothelial inflammation in naturally flow-disturbed LC of the aortic arch in mice. Treatment with rKLK10 in vivo (intravenous injection every 2 days for 5 days at 0.6 mg/kg) dramatically reduced VCAM1 expression in the *d-flow* (LC) region in the aortic arch of these mice (***Figure 2j, k***). We also observed a dose-dependent effect of rKLK10 on VCAM1 expression in the same study (***Figure 2—figure supplement 4***). Injection of rKLK10 at 0.6 mg/kg dose increased its plasma level to a peak of ~1600 ng/ml with a $t_{1/2}$ of 4.5 hr, becoming undetectable by 24 hr (***Figure 2—figure supplement 4***). These results demonstrate that either KLK10 overexpression using plasmids or rKLK10 treatment protects against EC inflammation both in vitro and in vivo under TNFα or *d-flow* conditions, whereas the reduction of KLK10 by *d-flow* condition or *KLK10* mRNA knockdown using siRNA increases inflammation.

Since NFκB is a well-known proinflammatory transcription factor, which induces expression of VCAM1 and ICAM1 and subsequent monocyte adhesion to ECs (***Baeriswyl et al., 2019***; ***Baeyens et al., 2014***; ***Chen et al., 2003***; ***Coleman et al., 2020***; ***Lay et al., 2019***; ***Mohan et al., 1997***; ***Petzold et al., 2009***; ***Stefanini et al., 2015***; ***Wang et al., 2009***; ***Wilson et al., 2013***), we tested whether KLK10 inhibits NFκB activation in response to shear stress and TNFα. We first found that KLK10 prevented phosphorylation (p-Ser536) and trans-nuclear location of p65, two important markers of NFκB activation, in response to TNFα (***Figure 3a–d***). KLK10 also prevented trans-nuclear location of p65 in response to acute shear challenge using LS condition (***Figure 3e, f***), which is well known to induce robust and transient NFκB activation (***Baeriswyl et al., 2019***; ***Baeyens et al., 2014***; ***Chen et al., 2003***; ***Coleman et al., 2020***; ***Lay et al., 2019***; ***Mohan et al., 1997***; ***Petzold et al., 2009***; ***Stefanini et al., 2015***; ***Wang et al., 2009***; ***Wilson et al., 2013***).

Next, we tested if rKLK10 treatment can protect the permeability barrier function of ECs. As a positive control, thrombin treatment increased the permeability of HAECs as measured by increased binding of fluorescently labeled (FITC)-avidin to biotin-gelatin as reported previously (***Dubrovskyi et al., 2013***). Overnight rKLK10 pretreatment prevented the permeability increase induced by thrombin in HAECs (***Figure 4a, b***). Similarly, rKLK10 reduced the permeability induced by OS (***Figure 4c, d***). Together, these results demonstrate the protective role of KLK10 in endothelial inflammation and barrier function.

## Treatment with rKLK10 inhibits atherosclerosis in *Apoe*$^{-/-}$ mice

Given its anti-inflammatory and barrier-protective effect in ECs, we tested if atherosclerosis development could be prevented by treating mice with rKLK10. For this study, we used the PCL model

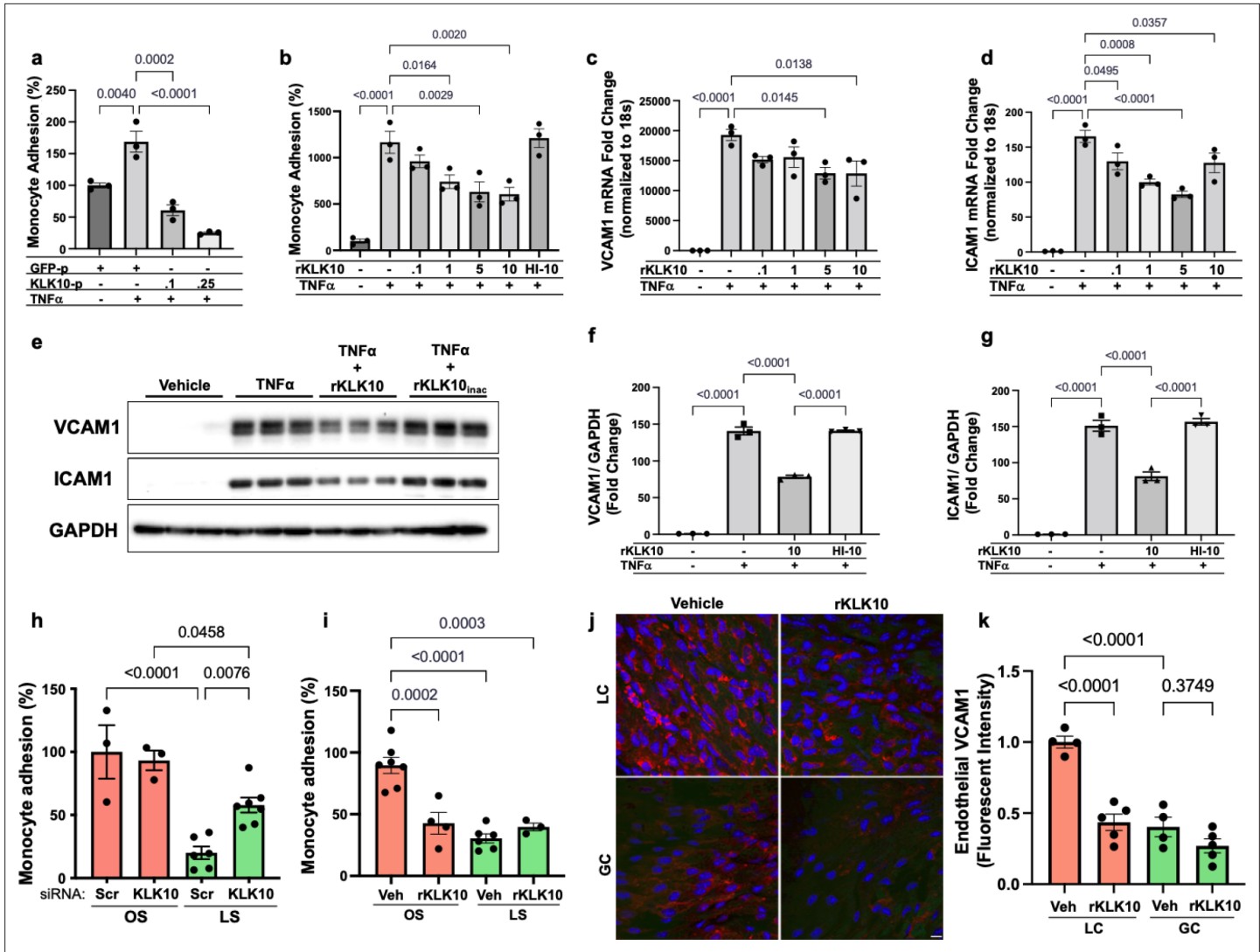

**Figure 2.** KLK10 inhibits inflammation in endothelial cells in vitro and in vivo. (**a**) THP-1 monocyte adhesion assay was carried out in human artery endothelial cells (HAECs) transfected with 0.1 or 0.25 μg of *KLK10* plasmid (KLK10-p) or GFP plasmid (GFP-p) for 48 hr followed by TNFα treatment (5 ng/ml for 4 hr). Data are represented as percentage of monocyte adhesion normalized to GFP-p control. *N* = 3. (**b**) THP-1 monocyte adhesion assay was carried out in HAECs treated with rKLK10 (0.1–10 ng/ml) or heat-inactivated rKLK10 (HI-10) for 24 hr followed by TNFα treatment (5 ng/ml for 4 hr). Data are represented as percentage of monocyte adhesion normalized to vehicle control. *N* = 3. (**c–g**) HAECs were treated with rKLK10 (0.1–10 ng/ml for 24 hr) followed by TNFα treatment (5 ng/ml for 4 hr) and expression of vascular cell adhesion molecule 1 (VCAM1) and intracellular adhesion molecule 1 (ICAM1) were assessed by quantitative real-time polymerase chain reaction (qPCR) (**c, d**) or western blot (**e–g**). *N* = 3. Data are represented as fold-change of the vehicle control and normalized to 18s or GAPDH (*Figure 2—source data 1*). (**h, i**) THP-1 monocyte adhesion assay was conducted on HAECs subjected to 24 hr of either laminar shear (LS; 15 dynes/cm²) or oscillatory shear (OS; ±5 dynes/cm²) with either (**h**) rKLK10 (100 ng/ml) or (**i**) *KLK10* siRNA (50 nM) or a nontargeting siRNA control. Data are represented as percentage of monocyte adhesion normalized to the control OS condition. *N* = 3–7. (**j**) C57BL/6J mice were injected with rKLK10 (0.6 mg/kg) or a vehicle control by tail vein once every 2 days for 5 days. The aortic arches were *en face* immunostained and imaged using confocal microscopy with an anti-VCAM1 antibody (red) and DAPI (blue). (**k**) Quantification of endothelial VCAM1 fluorescence intensity represented as fold-change normalized to control LC condition. *N* = 4–5. Scale bar = 10 μm. All data are represented as mean ± standard error of mean (SEM). Statistical analyses were performed using one-way analysis of variance (ANOVA) with Bonferroni correction for multiple comparisons.

The online version of this article includes the following source data and figure supplement(s) for figure 2:

**Source data 1.** Western blots for VCAM1, ICAM1, and GAPDH.

**Figure supplement 1.** KLK10 inhibits endothelial migration and tube formation, but not apoptosis or proliferation.

**Figure supplement 2.** KLK10 reduces inflammation in endothelial cell.

**Figure supplement 3.** Heat inactivation of rKLK10 prevents its anti-inflammatory effects on *VCAM1* and *ICAM1* mRNA expression.

*Figure 2 continued on next page*

*Figure 2 continued*

**Figure supplement 4.** rKLK10 inhibits vascular cell adhesion molecule 1 (VCAM1) expression in the *d-flow* region of the mouse aortic arch in a dose-dependent manner.

**Figure supplement 5.** Orthogonal projection of vascular cell adhesion molecule 1 (VCAM1) *en face* immunostaining and quantification.

**Figure supplement 6.** *KLK10* plasmid and *KLK10* siRNA overexpress and knockdown KLK10, respectively.

**Figure supplement 6—source data 1.** Western blots for HIS and GAPDH.

**Figure supplement 7.** Human rKLK10 purification from CHO cells.

**Figure supplement 7—source data 1.** rKLK10 coomassie blue gel.

of atherosclerosis to induce atherosclerosis rapidly in a flow-dependent manner in hyperlipidemic *Apoe*$^{-/-}$ mice fed with a high-fat diet. Injection with rKLK10 by tail vein (twice per week at 0.6 mg/kg for 3 weeks post-PCL surgery) significantly reduced atherosclerosis development and macrophage accumulation in the LCA (*Figure 5a–e*). The rKLK10 treatment showed no effect on plasma levels of total, LDL (low-density lipoprotein), and HDL (high-density lipoprotein) cholesterols and triglycerides (*Figure 5f–i*). Thus, rKLK10 showed an anti-atherogenic effect in vivo.

## Ultrasound-mediated overexpression of KLK10 inhibits atherosclerosis in *Apoe*$^{-/-}$ mice

We next asked whether overexpression of KLK10 using a plasmid vector could also inhibit atherosclerosis in vivo. For this study, we injected either KLK10 plasmid (pCMV-Igκ-*Klk10*-T2A-Luc) or luciferase plasmid (pCMV-Luc) as a control along with microbubbles to the hind-limbs of *Apoe*$^{-/-}$ mice and sonoporated the legs with ultrasound as previously described (*Liu et al., 2019*; *Borden et al., 2005*; *Shapiro et al., 2016*). The plasmid injection and sonoporation were repeated 10 days later to ensure sustained protein expression for the duration of the study. Bioluminescence imaging showed that all mice expressed luciferase in the hind-limbs at the conclusion of the study, indicating successful and sustained overexpression of the plasmids (*Figure 6a*).

Atherosclerotic plaque formation in the LCA was significantly reduced in the KLK10 overexpressing mice compared to the luciferase control (*Figure 6b*). Further assessment of the carotid artery sections by histochemical staining with hematoxylin and eosin (*Figure 6d, e*) showed decreased plaque area in the LCA of these mice. Circulating plasma KLK10 levels in the mice measured by ELISA at the time of sacrifice showed no measurable difference between the luciferase and KLK10 groups (*Figure 6—figure supplement 1*). This may be due to a waning plasmid expression at the sacrifice time. However, we found higher levels of KLK10 staining at the endothelial layer in the LCA and RCA (*Figure 6f, g*), as well as in the lung tissue samples as shown by western blot (*Figure 6h, i*). We observed no significant difference in plasma total cholesterol, triglycerides, HDLc, LDLc, and non-HDLc (*Figure 6j–n*) in the KLK10-injected mice compared to the control mice. These results demonstrate that treatment with KLK10 by either rKLK10 or KLK10 expression vector can inhibit atherosclerosis development in *Apoe*$^{-/-}$ mice.

## KLK10 expression is decreased in human coronary arteries with advanced atherosclerotic plaques

We next examined if KLK10 expression is altered in human coronary artery tissue sections with varying degrees of atherosclerotic plaques ($n = 40$ individuals, *Table 1*). KLK10 and CD31 immunostaining demonstrated that KLK10 expression was significantly reduced at the endothelial layer in arteries with significant plaques (grades 4–6; *Figure 7a, b*) than less-diseased arteries (grades 1–3).

## Discussion

Here, we describe that *s-flow* promotes, while *d-flow* inhibits, expression and secretion of KLK10 in ECs in vitro and in vivo. We found for the first time that KLK10 inhibits endothelial inflammation, endothelial barrier dysfunction, and reduces endothelial migration and tube formation, but not apoptosis or proliferation. Importantly, treatment of ECs in vitro with rKLK10 or a KLK10 expression plasmid inhibited endothelial inflammation induced by *d-flow* or TNFα. Moreover, treatment with rKLK10 or overexpression of KLK10 by ultrasound-mediated plasmid expression inhibited endothelial

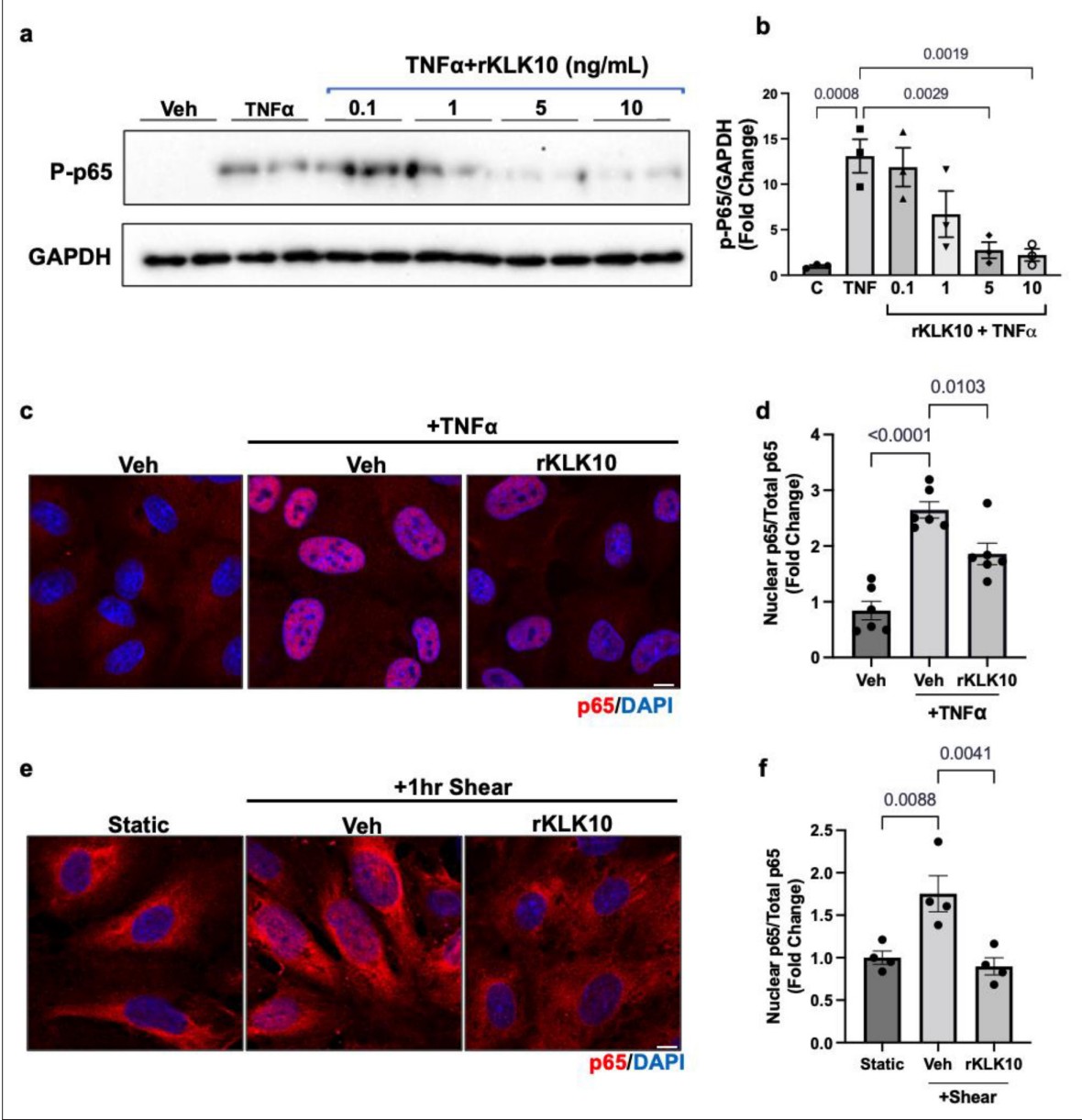

**Figure 3.** KLK10 inhibits NF $\kappa$ B p65 phosphorylation and nuclear translocation. (**a, b**) Human aortic endothelial cells (HAECs) were treated with rKLK10 (10 ng/ml) or vehicle for 16 hr followed by TNFα (5 ng/ml for 4 hr). Cell lysates were then collected and analyzed for phosphorylated p65 (p-p65) by sodium dodecyl sulfate–polyacrylamide gel electrophoresis (SDS–PAGE). Data are expressed as p-p65 fold-change normalized to GAPDH and vehicle control. $N$ = 3 (**Figure 3—source data 1**). (**c, d**) HAECs were treated with rKLK10 (10 ng/ml) or vehicle for 16 hr followed by TNFα (5 ng/ml for 4 hr). Cells were then fixed and immunostained for p65 using anti-p65 antibody. Data are expressed as nuclear p65/total p65, normalized to the vehicle control. $N$ = 6. (**e, f**) HAECs were treated with rKLK10 (10 ng/ml) or vehicle for 16 hr and exposed to shear for 1 hr. Cells were then fixed and immunostained for p65 using anti-p65 antibody. Data are expressed as nuclear p65/total p65, normalized to the static control. $N$ = 4. All data are represented as mean ± standard error of mean (SEM). Statistical analyses were performed using one-way analysis of variance (ANOVA) with Bonferroni correction.

The online version of this article includes the following source data for figure 3:

**Source data 1.** Western blots for p-p65 NFkB and GAPDH.

inflammation and atherosclerosis development in vivo. Our findings also indicate that KLK10 is likely to be important in human atherosclerotic plaque development. The protective effects of rKLK10 or plasmid-driven KLK10 expression on endothelial inflammation, barrier function, and atherosclerosis suggest its therapeutic potential for atherosclerosis treatment.

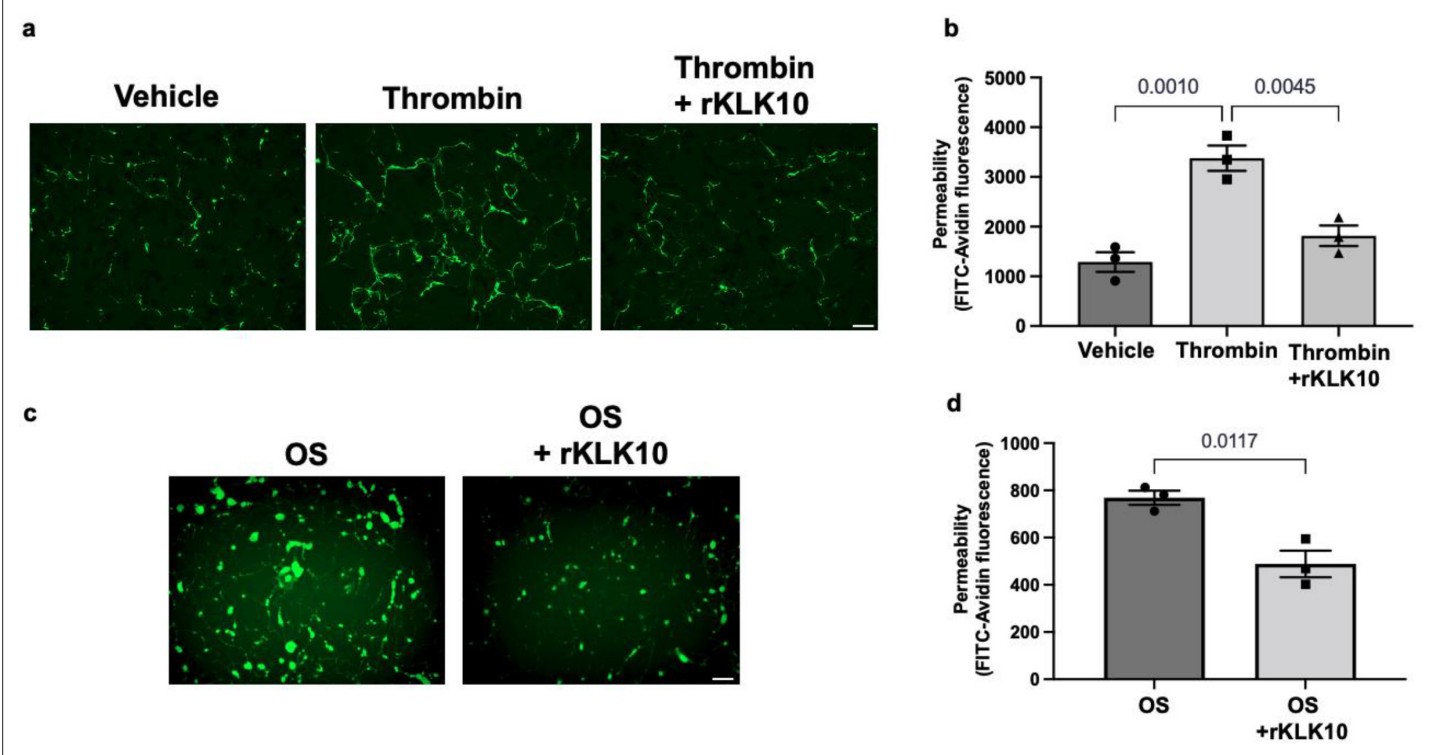

**Figure 4.** KLK10 protects endothelial permeability against thrombin and oscillatory shear (OS). Human aortic endothelial cells (HAECs) were grown to confluency on biotinylated gelatin and were treated with (**a, b**) rKLK10 (10 ng/ml) or vehicle for 16 hr followed by thrombin (5 U/ml for 30 min), or (**c, d**) exposed to OS (±5 dynes/cm²) with rKLK10 (10 ng/ml) or vehicle for 24 hr. Endothelial permeability was then measured by the binding of FITC-avidin to the biotinylated gelatin. (**b, d**) Quantification of endothelial permeability measured as FITC-avidin fluorescence intensity. N = 3 each. Scale bar = 50 μm. All data are represented as mean ± standard error of mean (SEM). Statistical analyses were performed using one-way analysis of variance (ANOVA) with Bonferroni correction (**b**) or paired *t*-test (**d**).

The *Klk10* mRNA transcript was primarily found in ECs, while KLK10 protein was found not only in ECs but also in the adventitia and subendothelial layer (*Figure 1*). It is important to note that our single-cell RNAseq and ATACseq analyses of *Klk10* expression in the mouse carotid artery clearly demonstrate that *Klk10* mRNA is highly expressed only in ECs but not in other cell types including the SMCs, fibroblasts, or immune cells (*Figure 1k, l*). In addition, KLK10 is a secreted protein, which could be released to the circulation to be found in other locations including the adventitia and diffuse to the subendothelial layer. Therefore, we conclude that KLK10 protein signals observed in the subendothelial and adventitial layers are likely to be originated from ECs.

*KLK10* expression is downregulated in breast, prostate, testicular, and lung cancer (*Goyal et al., 1998*; *Liu et al., 1996*; *Hu et al., 2015*; *Luo et al., 2001*; *Zhang et al., 2010*) but overexpressed in ovarian, pancreatic, and uterine cancer (*Luo et al., 2003*; *Yousef et al., 2005*; *Dorn et al., 2013*; *Tailor et al., 2018*). These suggest that abnormal, either too low or too high, levels of KLK10 are associated with various pathophysiological conditions. Overall, the effective concentration of rKLK10 we used in this study is within a reasonable range of human and mouse KLK10 levels in the plasma. Our mouse KLK10 ELISA study (*Figure 6—figure supplement 1*) showed that plasma KLK10 level in *Apoe*⁻/⁻ mice is in the range of 5–10 ng/ml. In humans, normal plasma KLK10 levels are ~0.5 ng/ml, with a range from nearly undetectable to ~20 ng/ml in various cancers patients (*Luo et al., 2003*; *Planque et al., 2008*). We found that KLK10 levels in HAECs exposed to the anti-inflammatory LS was ~0.3 ng/ml, which decreased to ~0.13 ng/ml by the proinflammatory OS (*Figure 1j*). In functional studies, we found that 1–10 ng/ml of rKLK10 inhibits permeability and inflammation in HAECs, which falls within the reasonable physiological range. The effective rKLK10 dose used in mouse studies was 0.6 mg/kg, although how this effective dose translates to humans will need to be further studied. It is also worth noting that the effect of rKLK10 on monocyte adhesion (*Figure 2b*) is weaker than that of KLK10 overexpression using the plasmid vector (*Figure 2a*). We speculate that KLK10 produced from

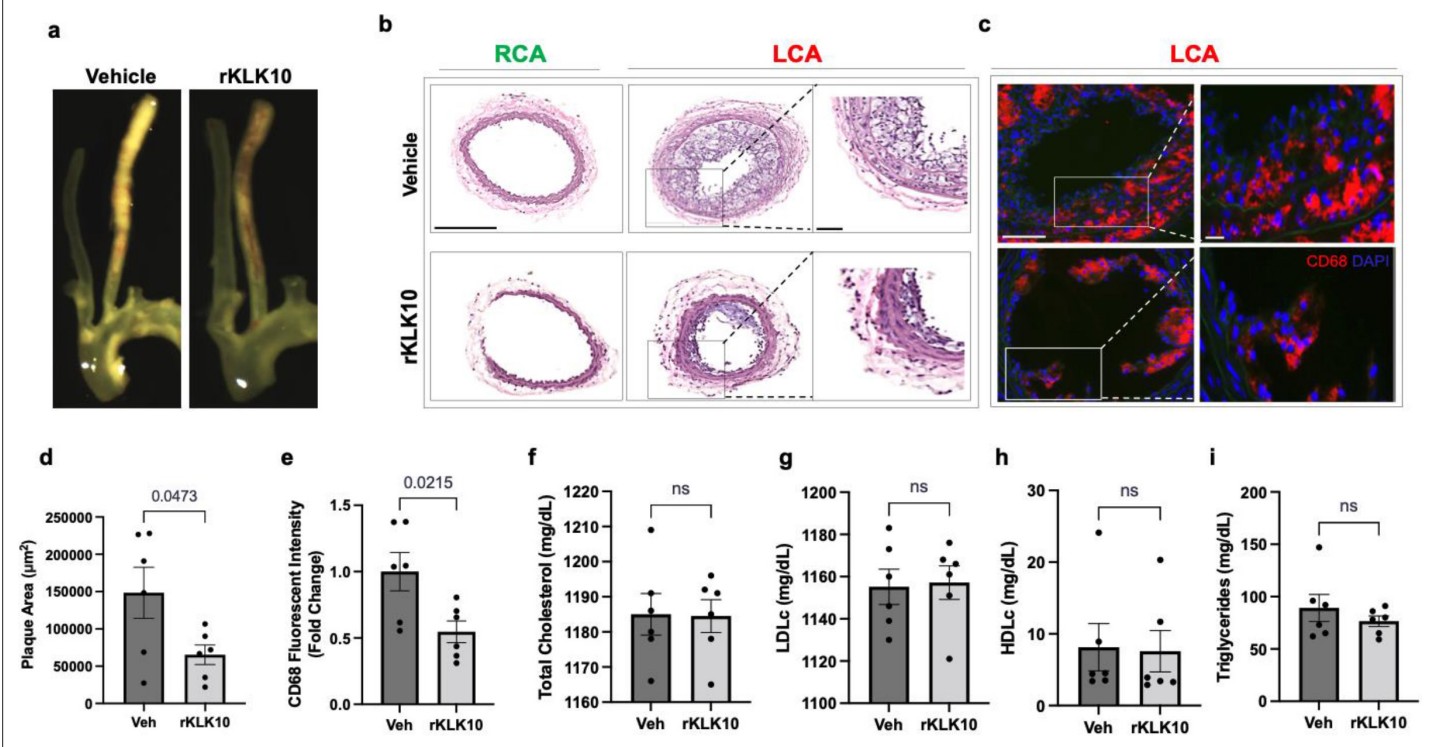

**Figure 5.** Treatment with rKLK10 inhibits atherosclerosis development in *Apoe*⁻/⁻ mice. (**a**) *Apoe*⁻/⁻ were subjected to partial carotid ligation and high-fat diet feeding. The mice received either rKLK10 (0.6 mg/kg) or vehicle injection every 3 days for the duration of 3 weeks. Left carotid artery (LCA) showed plaque development, which was reduced by rKLK10 as shown by dissection microscopy. Frozen sections from the LCA and right carotid artery (RCA) were stained with (**b**) H&E and (**c**) for CD68 in LCA. DAPI (blue). Scale bar low mag = 250 μm, high mag = 50 μm. (**d**) Plaque area was quantified from H&E staining and is represented as μm². (**e**) CD68 fluorescence intensity was quantified and is represented as the CD68 fold-change normalized to the control. Plasma lipid analysis of (**f**) total cholesterol, (**g**) low-density lipoprotein (LDL cholesterol), (**h**) high-density lipoprotein (HDL) cholesterol, or (**i**) triglycerides showed no effect of rKLK10 compared to control. All data are represented as mean ± standard error of mean (SEM). Statistical analyses were performed using paired *t*-test. *N* = 6. ns = not significant.

plasmid directly in HAECs is processed to be more effective than the rKLK10 produced and processed in CHO cells, which underwent multiple purification steps and storage conditions.

Interestingly, the anti-inflammatory effect of KLK10 seems to be unique in comparison to other KLKs expressed in ECs, including KLK8 and KLK11. Analysis of the sc-RNAseq dataset showed that *Klk8* and *Klk11* are two other KLK members expressed in ECs (***Figure 8—figure supplement 1***). We found that rKLK10, but not rKLK8 or rKLK11, inhibited endothelial inflammation in response to TNFα in HAECs (***Figure 8—figure supplement 2***).

Taken together, we demonstrated that KLK10 is a flow-sensitive protein that is upregulated by *s-flow* and downregulated by *d-flow* in ECs. Our results also demonstrate that KLK10 is a potent mediator of the anti-inflammatory, barrier-protective, and anti-atherogenic effects of *s-flow* in an autocrine manner in ECs (***Figure 8***). KLK10 may serve as potential anti-atherogenic therapeutic targets.

## Materials and methods

### Key resources table

| Reagent type (species) or resource | Designation | Source or reference | Identifiers | Additional information |
|---|---|---|---|---|
| Gene (human) | *KLK10* | | Gene ID: 5655 | Kallikrein-related peptidase 10 |
| Gene (mouse) | *Klk10* | | Gene ID: 69,540 | Kallikrein-related peptidase 10 |
| Strain, strain background (mouse) | C57BL/6J | The Jackson Laboratory | 000664 | Male, 6–10 weeks of age |

*Continued on next page*

*Continued*

| Reagent type (species) or resource | Designation | Source or reference | Identifiers | Additional information |
|---|---|---|---|---|
| Genetic reagent (mouse) | *Apoe*−/− (B6.129P2-*Apoe*tm1Unc/J) | The Jackson Laboratory | 002052 | C57BL/6J Mice homozygous for the *Apoe*tm1Unc mutation. Male, 6–10 weeks of age |
| Peptide, recombinant protein | Recombinant human KLK10 | RayBiotech | 230-00040-10 | Human produced in *E. coli* |
| Peptide, recombinant protein | Recombinant human KLK10-6xHis | This paper | Gene ID: 5655 | Ala34-Asn276 with C-terminal His tag Human produced in CHO cells |
| Peptide, recombinant protein | Human TNFα | Thermo Fisher | PHC3011 | Human produced in *E. coli* |
| Recombinant DNA reagent | pcDNA3.4_h*KLK10*-6X His Plasmid | This paper | Gene ID: 5655 | Human *KLK10* Met1- Asn276 with C-terminal His tag |
| Recombinant DNA reagent | pCMV-Igκ-*Klk10*-T2A-Luc Plasmid | This paper | Gene ID: 69,540 | Mouse *Klk10* Met1-Lys278 with secretion tag and cleavable Luc reporter |
| Recombinant DNA reagent | PCMV-Luciferase Plasmid | Addgene | #45,968 | |
| Recombinant DNA reagent | PmaxGFP Plasmid | Lonza | #D-00059 | |
| Recombinant DNA reagent | Human *KLK10* siRNA | Dharmacon | J-005907-08 | |
| Recombinant DNA reagent | Human Scrambled siRNA | Dharmacon | D-001810-10-05 | |
| Cell line | Primary Human Aortic Endothelial Cells | Cell Applications | 304-05a | 25–40-Year-old males. Multiple lots. Cell identity confirmed through diacetylated LDL and FACs. Company tested cells free of mycoplasma, bacteria, yeast, and fungi |
| Cell line | Primary Human Umbilical Vein Endothelial Cells | Lonza | CC-2519 | Pooled female donors. Multiple lots. Cell identity confirmed through diacetylated LDL and FACs. Company tested cells free of mycoplasma, bacteria, yeast, and fungi |
| Cell line | THP1 Human Monocytes | ATCC | Cat TIB-202 | STR Profiling and mycoplasma testing done by ATCC |
| Biological sample (human) | Human Coronary Arteries | Lifelink Georgia | | Deidentified human hearts not suitable for cardiac transplantation donated to LifeLink of Georgia |
| Antibody | KLK10 (Rabbit Polyclonal) | Bioss | Bioss Cat# bs-2531R, RRID:AB_10882440 | IF (1:100), WB (1:1000) |
| Antibody | CD31 (Rabbit Polyclonal) | Abcam | Abcam Cat# ab28364, RRID:AB_726362 | IF (1:100) |
| Antibody | VCAM1 (Rabbit Monoclonal) | Abcam | Abcam Cat# ab134047, RRID:AB_2721053 | IF (1:100) WB (1:1000) |
| Antibody | VE-Cadherin (Mouse monoclonal) | Santacruz | Santa Cruz Biotechnology Cat# sc-9989, RRID:AB_2077957 | IF (1:100) |
| Antibody | NFKB P65 (Rabbit Monoclonal) | Cell Signaling | Cell Signaling Technology Cat# 8242, RRID:AB_10859369 | IF (1:100) |
| Antibody | phospho-NFκB p65 S356 (Rabbit Monoclonal) | Cell Signaling | Cell Signaling Technology Cat# 3033, RRID:AB_331284 | WB (1:1000) |
| Antibody | Alexa Fluor Secondaries | Thermo Fisher | | IF (1:500) |
| Antibody | Ki67 (Rabbit Polyclonal) | Abcam | Abcam Cat# ab15580, RRID:AB_443209 | IF (1:100) |
| Commercial assay or kit | H&E Staining Kit | American Mastertech | KTHNEPT | |
| Commercial assay or kit | Human *KLK10* ELISA | MyBioSource | Cat. #: MBS009286 | |

*Continued on next page*

*Continued*

| Reagent type (species) or resource | Designation | Source or reference | Identifiers | Additional information |
|---|---|---|---|---|
| Commercial assay or kit | Mouse *Klk10* ELISA | NovateinBio | BG-MUS11429 | |
| Commercial assay or kit | TUNEL Staining Kit | Roche | 12156792910 | |
| Chemical compound | 2′,7′-bis(carboxyethyl)-5 (6)-carboxyfluorescein-AM | Thermo Fisher | B1150 | 1 mg/ml |
| Primers | Human *KLK10* | | | For: GAGTGTGAGGTCTTCTACCCTG Rev: ATGCCTTGGAGGGTCTCGTCAC |
| Primers | Mouse *Klk10* | | | For: CGC TAC TGA TGG TGC AAC TCT Rev: ATA GTC ACG CTC GCA CTG G |
| Primers | Human/Mouse 18s | | | For: AGGAATTGACGGAAGGGCACCA Rev: GTGCAGCCCCGGACATCTAAG |
| Primers | Human *VCAM1* | | | For: GATTCTGTGCCCACAGTAAGGC Rev: TGGTCACAGAGCCACCTTCTTG |
| Primers | Human *ICAM1* | | | For: AGCGGCTGACGTGTGCAGTAAT Rev: TCTGAGACCTCTGGCTTCGTCA |
| Software, algorithm | Zen Blue | Zeiss | | Confocal Microscopy |
| Software, algorithm | ImageJ | NIH | | Image Analysis |
| Other | DAPI Mounting media | Vector Biolabs | H-1200-10 | Methods – immunostaining of mouse artery sections an *en face* preparation of the aortic arch |
| Other | Oligofectamine | Thermo Fisher | 12252011 | Methods – overexpression or knockdown experiments in vitro |
| Other | Lipofectamine | Thermo Fisher | L3000008 | Methods – overexpression or knockdown experiments in vitro |

## PCL surgery

All animal studies were performed with male C57BL/6J or *Apoe*$^{-/-}$ mice (Jackson Laboratory), were approved by Institutional Animal Care and Use Committee by Emory University, and were performed in accordance with the established guidelines and regulations consistent with federal assurance. All studies using mice were carried out with male mice at 6–10 weeks to reduce the sex-dependent variables. For PCL studies, mice at 10 weeks were anesthetized and three of four caudal branches of LCA (left external carotid, internal carotid, and occipital artery) were ligated with 6–0 silk suture, but the superior thyroid artery was left intact. The development of *d-flow* with characteristic low and oscillating shear stress in each mouse was determined by ultrasound measurements as we described (*Nam et al., 2009*). Following the partial ligation, mice were either continued to be fed chow-diet for 2 days or high-fat diet for atherosclerosis studies for 3 weeks as specified in each study.

Endothelial-enriched RNA was prepared from the LCA and the contralateral RCA control following 48 hr after the partial ligation as we described previously (*Nam et al., 2009*).

## Immunostaining of mouse artery sections and *en face* preparation of the aortic arch

For mouse frozen section staining studies, fresh mouse aortas were fixed in 4% paraformaldehyde for 15 min and placed in Tissue-Tek OCT compound, snap-frozen in liquid nitrogen, and sectioned at 7 µm as we described (*Son et al., 2013*). Sections were then permeabilized using 0.1% Triton X-100 in Phosphate-buffered saline (PBS) for 15 min, blocked for 2 hr with 10% donkey serum, and incubated with anti-KLK10 (BiossUSA bs-2531R, 1:100) or anti-CD31 (R&D Systems AF3628, 1:100) primary antibodies overnight at 4°C followed by Alexa Fluor secondary antibodies (Thermo Fisher

Scientific, 1:500) for 2 hr at room temperature. All images were taken with a Zeiss (Jena, Germany) LSM800 confocal microscope. Endothelial KLK10 fluorescent intensity was measured with NIH ImageJ using CD31 as a reference. Hematoxylin and eosin staining (American Mastertech) and plaque area quantification were performed using ImageJ software (NIH) as we described (*Chang et al., 2007*; *Kim et al., 2013*).

For *en face* immunostaining, mice were euthanized under $CO_2$ and the aortas were pressure fixed with 10% formalin saline (*Nam et al., 2009*). The aortas were carefully cleaned in situ, and the aortic arches and thoracic aortas were dissected, opened longitudinally, and fixed in 4% paraformaldehyde for 1 hr, permeabilized using 0.1% Triton X-100 in PBS for 15 min, blocked for 2 hr with 10% donkey serum, and incubated with anti-KLK10 (BiossUSA bs-2531R, 1:100), anti-VCAM1 (Abcam ab134047, 1:100), or anti-VE-Cadherin (Santa Cruz sc-9989, 1:100) primary antibodies overnight at 4°C followed by Alexa Fluor-647 secondary antibodies (Thermo Fisher Scientific, 1:500) for 2 hr at room temperature. The LC and GC of each arch were separated and the aortas were then mounted on glass slides with VectaShield that contained DAPI (Vector Laboratories). *En face* images were collected as a Z-stack with a Zeiss LSM 800 confocal microscope. We used three Z-sections showing the endothelial layer using the internal elastic laminar as a reference from each tissue sample to quantify VCAM1 or KLK10 expression in the ECs (orthogonal image shown in *Figure 2—figure supplement 5*). The fluorescence intensity was quantified using the NIH ImageJ program.

## Cell culture and in vitro shear stress study

HAECs were obtained from Lonza and maintained in EGM2 medium (Lonza) supplemented with 10% fetal bovine serum (Hyclone), 1% bovine brain extract, 10 mM L-glutamine, 1 µg/ml hydrocortisone hemisuccinate, 50 µg/ml ascorbic acid, 5 ng/ml EGF, 5 ng/ml VEGF, 5 ng/ml FGF, and 15 ng/ml IGF-1 as we described (*Son et al., 2013*). HUVECs were purchased from BD Biosciences, cultured in M199 media (Cellgro) supplemented with 20% fetal bovine serum (Hyclone), 1% bovine brain extract, 10 mM L-glutamine, and 0.75 U/ml heparin sulfate as we described (*Ni et al., 2011*). All ECs were grown at 5% $CO_2$ and 37°C and used between passages *5* and *9*. EC identity was confirmed through diacetylated-LDL uptake and FACs-based cell sorting. THP-1 monocytes were obtained from ATCC and maintained in RPMI-1640 medium supplemented with 10% fetal bovine serum and 0.05 mM 2-mercaptoethanol at 5% CO2 and 37°C as we described (*Ni et al., 2011*). THP-1 STR Profiling and mycoplasma testing were done by ATCC. For flow experiments, confluent HAECs or HUVECs were exposed to steady unidirectional laminar shear stress (LS, 15 dynes/cm$^2$) or bidirectional oscillatory shear stress (OS, ±5 dynes/cm$^2$ at 1 Hz), mimicking *s-flow* and *d-flow* conditions, respectively, using the cone-and-plate viscometer for 24 hr experiments, as we reported (*Jo et al., 2006*; *Chang et al., 2007*).

## Preparation of whole-cell lysate and immunoblotting

After treatment, cells were washed 3× with ice-cold Hank's Balanced Salt Solution (HBSS) and lysed with Radioimmunoprecipitation Assay buffer (RIPA) buffer containing protease inhibitors (Boston Bioproducts BP-421) *Son et al., 2013*. The protein content of each sample was determined by Pierce BCA protein assay. Aliquots of cell lysate were resolved on 10% to 12% sodium dodecyl sulfate–polyacrylamide gels and subsequently transferred to a polyvinylidene difluoride membrane (Millipore). The membrane was incubated with the following primary antibodies: anti-KLK10 (BiossUSA bs-2531R, 1:1000), anti-GAPDH (Abcam ab23565, 1:2000), anti-β-actin (Sigma-Aldrich A5316, 1:2000), anti-VCAM1 (Abcam ab134047, 1:1000), anti-ICAM1 (Abcam ab53013, 1:1000), and anti-phospho-NFκB p65 S356 (Cell Signaling #3033, 1:1000) overnight at 4°C in 5% milk in TBST at the concentration recommended by the manufacturer, followed by secondary antibody addition for 1 hr at RT in 5% milk in TBST. Protein expression was detected by a chemiluminescence method (*Son et al., 2013*).

## Quantitative real-time polymerase chain reaction

Total RNAs were isolated using RNeasy Mini Kit (Qiagen 74106) and reverse transcribed to cDNA using High-Capacity cDNA Reverse Transcription Kit (Applied Biosystems 4368814). qPCR was performed for genes of interests using VeriQuest Fast SYBR QPCR Master Mix (Affymetrix 75690) with custom designed primers (*Table 2*) using 18S as house-keeping control as we previously described (*Son et al., 2013*).

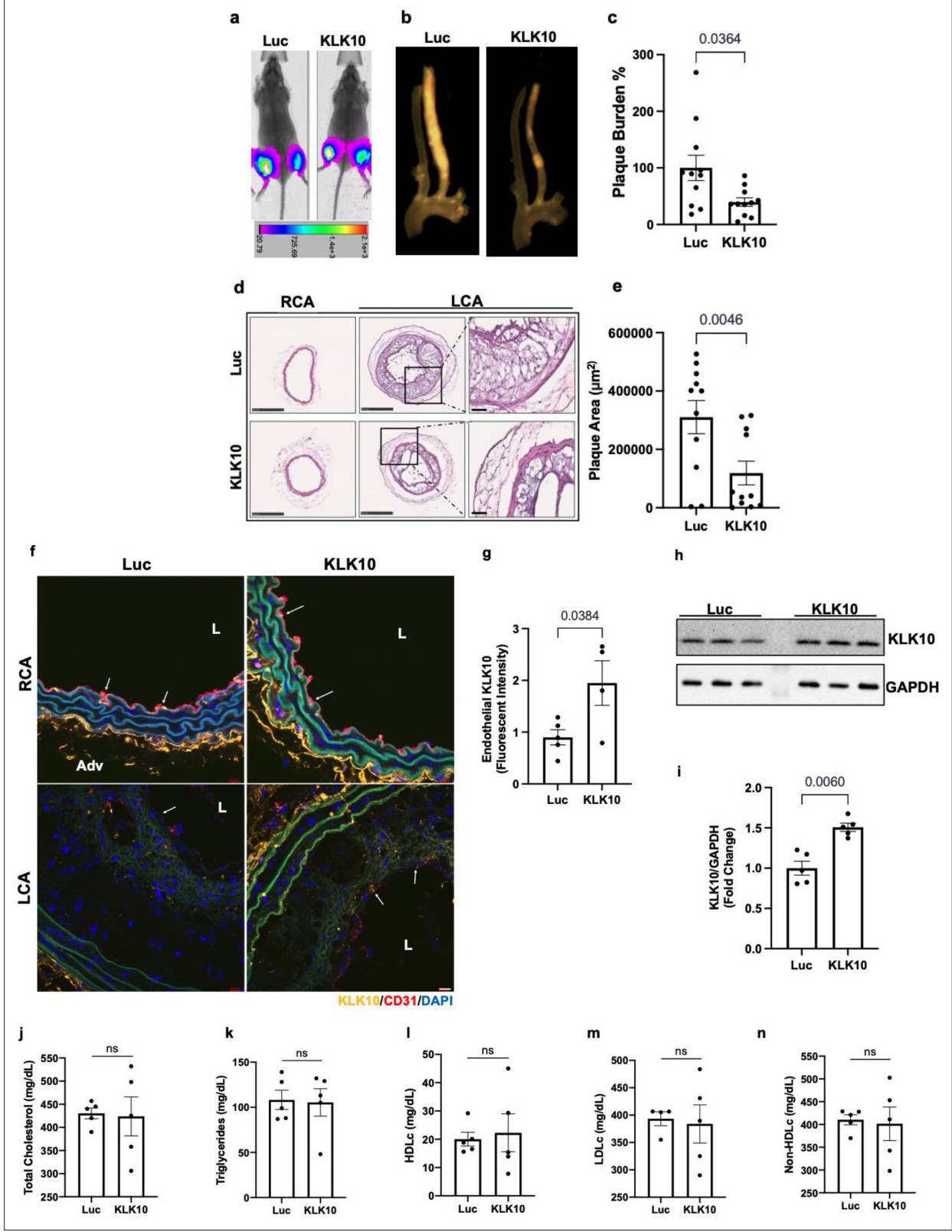

**Figure 6.** Ultrasound-mediated overexpression of *Klk10* plasmid inhibits atherosclerosis development. (**a**) Bioluminescent imaging of *Apoe⁻/⁻* partial carotid ligation (PCL) mice on a high-fat diet injected with luciferase control plasmid or *Klk10*-luciferase plasmid, measured in photons/second. (**b**) Gross plaque images of excised carotid arteries and (**c**) quantification of plaque burden normalized to the percentage of the luciferase control. (**d**) H&E staining of sections from the left carotid artery (LCA) and right carotid artery (RCA) of mice injected with luciferase control plasmid or *Klk10*-luciferase

*Figure 6 continued on next page*

*Figure 6 continued*

plasmid. Scale bar low mag = 250 µm, high mag = 50 µm. (**e**) Quantification of plaque area measured in µm². All data are represented as mean ± standard error of mean (SEM). Statistical analyses were performed using paired *t*-test. N = 11. (**f**) Sections from the RCA and LCA were coimmunostained with anti-KLK10 (orange) and anti-CD31 (red) antibodies. Blue is DAPI. Arrows indicate the ECs. L is the lumen and Adv is the adventitia. Scale bar = 10 µm. (**g**) Quantification of endothelial KLK10 fluorescent intensity represented as fold-change normalized to luciferase control. (**h**) Western blot analysis of KLK10 expression in lung tissue from mice injected with control luciferase plasmid or *Klk10* plasmid (*Figure 6—source data 1*). (**i**) Quantification of KLK10 expression normalized to GAPDH and luciferase control. Plasma lipid analysis of (**j**) total cholesterol, (**k**) triglycerides, (**l**) high-density lipoprotein (HDL) cholesterol, (**m**) low-density lipoprotein (LDL) cholesterol, or (**n**) non-HDL cholesterol. All data are represented as mean ± SEM. Statistical analyses were performed using paired *t*-test. N = 5. ns = not significant.

The online version of this article includes the following source data and figure supplement(s) for figure 6:

**Source data 1.** Western blots for KLK10 and GAPDH.

**Figure supplement 1.** KLK10 level in the mouse plasma.

## KLK10 ELISAs

KLK10 secreted into the conditioned cell culture media from HAECs exposed to shear stress was measured by using a human KLK10 ELISA kit (MyBioSource, MBS009286). KLK10 in mouse plasma was measured by using a mouse KLK10 ELISA kit (NovateinBio, BG-MUS11429).

## rKLK10 and KLK10 plasmids

Initially, human rKLK10 (Ala34-Asn276 with a 6× N-terminal His tag) produced in *E. coli* (Ray Biotech, 230-00040-10) was used. Additional studies using human rKLK10 produced in the mammalian CHO-K1 cells validated the initial results. Most studies were carried out using human rKLK10 produced in CHO-K1 cells using a full-length expression vector (pcDNA3.4 h*KLK10*-6X His). rKLK10 with a 6× C-terminal His tag was affinity purified using HisPur Ni-NTA Resin (Thermo Scientific) per the manufacturer's instruction (*Figure 2—figure supplement 7*) using the conditioned medium. Amino acid sequencing analysis of the purified rKLK10 by mass spectrometry showed that our rKLK10 preparation was a mature form expressing Ala34-Asn276 (data not shown).

## Overexpression or knockdown experiments in vitro

Cells were transiently transfected with a human *KLK10*-encoding plasmid (pcDNA3.4 h*KLK10*-6X His) at 0.1–2 µg/ml or as a control a GFP plasmid (PmaxGFP, Lonza, Cat. No. D-00059) using Lipofectamine 3000 (Invitrogen, Cat. No. L3000008) as we described (*Son et al., 2013*). Alternatively, cells were transfected with *KLK10* siRNA (Dharmacon; J-005907-08) or Scrambled siRNA (Dharmacon; D-001810-10-05) using Oligofectamine (Invitrogen, Cat. No. 12252011) as we described (*Son et al., 2013*). Overexpression and knockdown of KLK10 were confirmed in HAECs (*Figure 2—figure supplement 6*).

**Table 1.** Patient characteristics.

|  | Age (mean ± SEM) | Stroke | Hypertension | Diabetes | Smoking |
|---|---|---|---|---|---|
| Total (*n* = 40) | 52.25 ± 13.35 | 15 | 26 | 8 | 17 |
| Sex |  |  |  |  |  |
| Male (*n* = 27) | 53.21 ± 11.15 | 10 | 16 | 5 | 12 |
| Female (*n* = 13) | 51.93 ± 16.22 | 5 | 10 | 3 | 5 |
| Race |  |  |  |  |  |
| White (*n* = 21) | 54.95 ± 13.14 | 8 | 12 | 3 | 9 |
| Black (*n* = 17) | 49.72 ± 14.17 | 7 | 14 | 5 | 7 |
| Hispanic (*n* = 2) | 44 ± 5.66 | 0 | 0 | 0 | 1 |

SEM, standard error of mean.

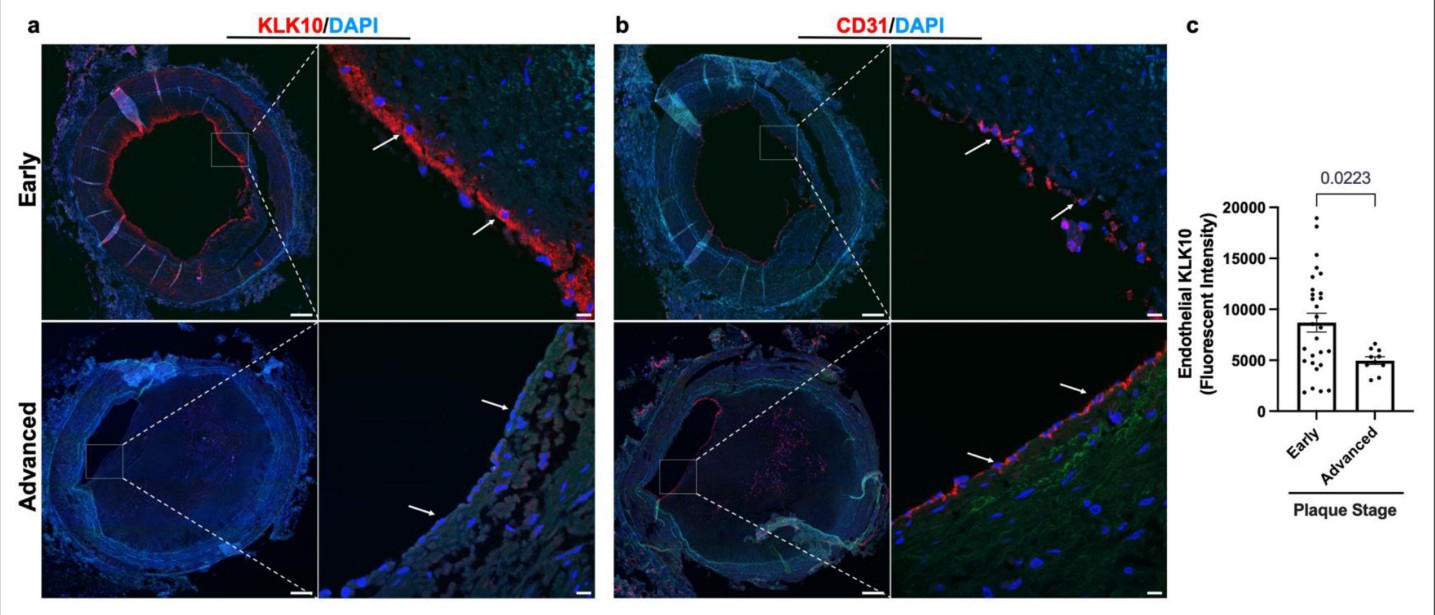

**Figure 7.** KLK10 expression is decreased in human coronary arteries with advanced atherosclerotic plaques. (**a**) Human coronary artery sections with varying degrees of atherosclerotic lesions were stained with anti-KLK10 antibody (red) and DAPI (blue). Scale bar low mag = 500 µm, scale bar; high mag = 50 µm. Arrows indicate endothelial cells. (**b**) Consecutive arterial sections from the same patients were stained with anti-CD31 antibody (red) and DAPI (blue). (**c**) Quantification of endothelial KLK10 fluorescence intensity in lower stage plaques (AHA grades 1–3) and advanced stage plaques (AHA grades 4–6). Data are from 40 different patients. Statistical analyses were performed using unpaired *t*-test. Mean ± standard error of mean (SEM) (*Table 1*).

## Endothelial functional assays

Endothelial migration was measured by the endothelial scratch assay, as we described (*Tressel et al., 2007*). Briefly, HUVECs were treated with rKLK10 at increasing doses overnight and cell monolayers were scratched with a 200 µl pipette tip. The monolayer was washed once, and the medium was replaced with 2% serum media. After 6 hr, the number of cells migrated into the scratch area was quantified microscopically using NIH ImageJ.

Endothelial apoptosis was determined using the TUNEL apoptosis assay, as we described (*Alberts-Grill et al., 2012*). Briefly, HUVECs were treated with rKLK10 at increasing doses overnight and the cells were fixed using 4% paraformaldehyde for 15 min and permeabilized with 0.1% Triton X-100 for 15 min. TUNEL staining was then performed using a commercially available kit (Roche, 12156792910) and the number of TUNEL-positive cells was counted using NIH ImageJ.

Endothelial proliferation was determined using Ki67 immunohistochemistry, as we described (*Wang et al., 2019*). Briefly, HUVECs were treated with rKLK10 at increasing doses overnight and the cells were washed twice with PBS, fixed using 4% paraformaldehyde for 15 min, and permeabilized with 0.1% Triton X-100 for 15 min. After blocking with 10% Goat Serum for 2 hr at RT, cells were incubated overnight at 4°C with rabbit anti-Ki67 primary antibody (Abcam ab15580, 1:100). The following day, cells were washed three times with PBS, incubated for 2 hr at RT protected from light with Alexa Fluor-647-labeled goat anti-rabbit IgG (1:500 dilution), and counterstained with VectaShield that contained DAPI (Vector Laboratories). The number of Ki67-positive cells was counted using NIH ImageJ.

Endothelial tube formation was measured using a Matrigel tube formation assay, as we described (*Tressel et al., 2007*). Briefly, HUVECs were seeded in a growth factor reduced Matrigel (BD Bioscience) coated 96-well plate and incubated with rKLK10 (100 ng/ml) for 6 hr at 37°C. Tubule formation was quantified microscopically by measuring tubule length using NIH ImageJ.

Endothelial permeability was determined by FITC-avidin binding to biotinylated gel, as previously described (*Dubrovskyi et al., 2013*). Briefly, HAECs were seeded on biotinylated gelatin and treated with rKLK10 overnight followed by thrombin (5 U/ml) for 4 hr or OS for 24 hr as described above. Following the completion of the experiments, FITC-avidin was added to the cells and fluorescent intensity was measured using NIH ImageJ.

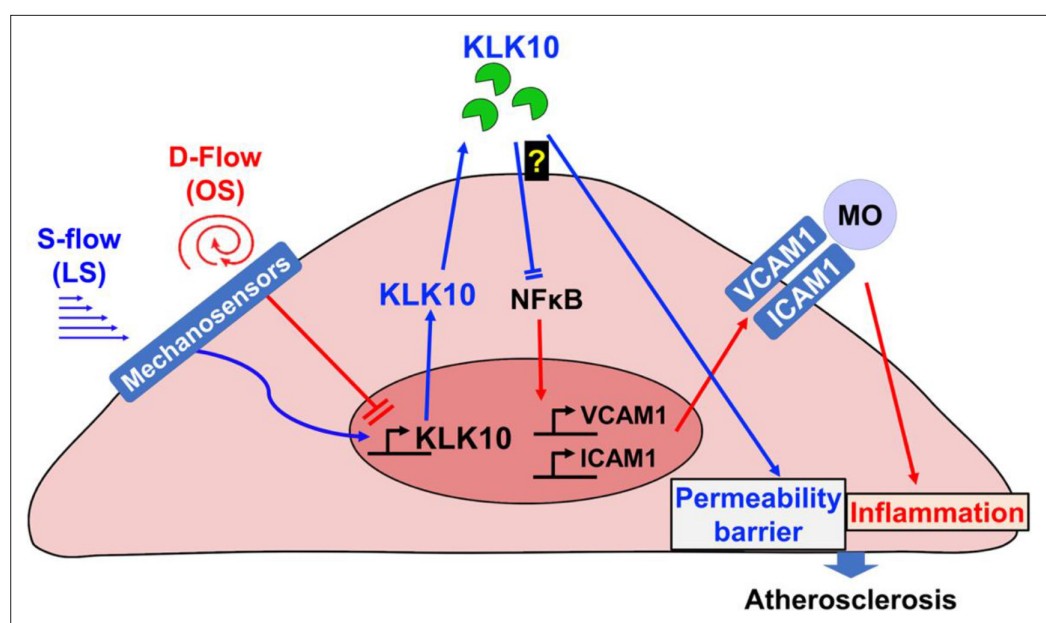

**Figure 8.** Flow-sensitive KLK10 inhibits endothelial inflammation and protects permeability barrier, ultimately reducing atherosclerosis. KL10 is upregulated by *s-flow* and downregulated by *d-flow* at the genomic and protein levels. Under *s-flow* conditions when KLK10 is expression is high, KLK10 inhibits NF $\kappa$ B and expression of vascular cell adhesion molecule 1 (VCAM1) and intracellular adhesion molecule 1 (ICAM1), thereby preventing monocyte adhesion. Additionally, KLK10 produced by *s-flow* protects the endothelial permeability barrier. Together, the anti-inflammatory and barrier-protective effects of KLK10 lead to an overall protection against atherosclerosis.

The online version of this article includes the following source data and figure supplement(s) for figure 8:

**Figure supplement 1.** Single-cell RNA sequencing (scRNAseq) analysis of *Klk8* and *Klk11* from the partial carotid ligation (PCL) mouse model.

**Figure supplement 2.** rKLK10, but not rKLK8 or rKLK11, inhibits monocyte adhesion and vascular cell adhesion molecule 1 (VCAM1) expression in human aortic endothelial cells (HAECs) exposed to TNFα.

**Figure supplement 2—source data 1.** Western blots for VCAM1 and beta-actin.

**Table 2.** Quantitative real-time polymerase chain reaction (qPCR) primers.

| Primer (custom) | Sequence |
|---|---|
| h_*KLK10* For | GAGTGTGAGGTCTTCTACCCTG |
| h_*KLK10* Rev | ATGCCTTGGAGGGTCTCGTCAC |
| m_*Klk10* For | CGC TAC TGA TGG TGC AAC TCT |
| m_*Klk10* Rev | ATA GTC ACG CTC GCA CTG G |
| H/M *18*S For | AGGAATTGACGGAAGGGCACCA |
| H/M *18*S Rev | GTGCAGCCCCGGACATCTAAG |
| h_*VCAM1* For | GATTCTGTGCCCACAGTAAGGC |
| h_*VCAM1* Rev | TGGTCACAGAGCCACCTTCTTG |
| h_*ICAM1* For | AGCGGCTGACGTGTGCAGTAAT |
| h_*ICAM1* Rev | TCTGAGACCTCTGGCTTCGTCA |

Monocyte adhesion to ECs was determined using THP-1 monocytes (ATCC TIB-202) as we described (*Son et al., 2013*). In brief, THP-1 cells ($1.5 \times 10^5$ cells/ml) were labeled with a fluorescent dye 2',7'-bis(carboxyethyl)-5 (6)-carboxyfluorescein-AM (Thermo Fisher Scientific B1150; 1 mg/ml) in serum-free RPMI medium (Thermo Fisher Scientific 11875093) for 45 min at 37°C. After exposure to flow or other experimental treatments, the ECs were washed in RPMI medium before adding 2',7'-bis(carboxyethyl)-5 (6)-carboxyfluorescein-AM-loaded THP-1 cells. After a 30-min incubation at 37°C under no-flow conditions, unbound monocytes were removed by washing the endothelial dishes 5× with HBSS and cells with bound monocytes were fixed with 4% paraformaldehyde for 10 min. Bound monocytes were quantified by counting the number of labeled cells at the endothelium under a fluorescent microscope.

NFκB p65 nuclear translocation was performed using HAECs treated with rKLK10 (10 ng/ml for

16 hr) followed by TNFα (5 ng/ml for 4 hr) or LS (20 dynes/cm$^2$ for 1 hr). Cells were washed three times, fixed with 4% paraformaldehyde for 15 min, and then permeabilized using 0.1% Triton X-100 in PBS for 15 min. Cells were then blocked for 2 hr with 10% donkey serum, and incubated with anti-p65 antibody (Cell Signaling #8242) overnight at 4°C followed by Alexa Fluor secondary antibodies (Thermo Fisher Scientific, 1:500) for 2 hr at room temperature. All images were taken with a Zeiss (Jena, Germany) LSM800 confocal microscope and nuclear p65 fluorescence intensity was quantified in comparison to total p65 fluorescence intensity using NIH ImageJ.

## rKLK10 treatment and KLK10 overexpression in C57BL/6J and *Apoe*$^{-/-}$ mice

Two independent methods were used, rKLK10 and *Klk10* plasmid, to treat mice with KLK10. Treatment with rKLK10 was first performed in C57BL/6J mice by administering rKLK10 (0.006–0.6 mg/kg) by tail vein once every 2 days and sacrificed on day 5. At the completion of the study, mice were euthanized by CO$_2$ inhalation and *en face* preparation of the aorta was performed as we described (*Son et al., 2013*). Alternatively, *Apoe*$^{-/-}$ on a high-fat diet containing 1.25% cholesterol, 15% fat, and 0.5% cholic acid were given the PCL surgery and rKLK10 or vehicle was administered by tail vein once every 3 days for 3 weeks as we described (*Son et al., 2013*). Following the completion of the study, mice were euthanized by CO$_2$ inhalation and the aortas were excised, imaged, and sectioned for IHC (*Son et al., 2013*).

*Klk10* plasmid overexpression was performed using ultrasound-mediated sonoporation method of gene therapy as reported (*Liu et al., 2019*; *Borden et al., 2005*; *Shapiro et al., 2016*). Briefly, perfluoropropane microbubbles encapsulated by DSPC and DSPE-PEG2000 (9:1 molar ratio) were made using the shaking method as previously described (*Liu et al., 2019*; *Borden et al., 2005*; *Shapiro et al., 2016*). *Klk10* plasmid expressing secreted KLK10 and luciferase (pCMV-Igκ-*Klk10*-T2A-Luc) from GENEWIZ or luciferase plasmid (pCMV-Luc) from Invitrogen (50 µg each) was then mixed with the microbubbles (5 × 10$^5$) and saline to reach 20 µl total volume. Following PCL, *Apoe*$^{-/-}$ mice were intramuscular injected to the hind-limbs with the plasmid-microbubble solution. The injected areas of the hind-legs were then exposed to ultrasound (0.35 W/cm$^2$) for 1 min, and repeated 10 days later. At the completion of the study 3 weeks after the partial ligation and on high-fat diet, mice were anesthetized, administered with luciferin (IP; 3.75 mg) and imaged for bioluminescence on a Bruker In Vivo Xtreme X-ray Imaging System. Mice were then euthanized by CO$_2$ inhalation and the aortas were excised, imaged, and sectioned for staining as described above.

## Immunohistochemical staining of sections from human coronaries

For human coronaries arteries, 2 mm cross-sections of the left anterior descending arteries were obtained from deidentified human hearts not suitable for cardiac transplantation donated to LifeLink of Georgia. The deidentified donor information is shown in *Table 1*. Tissues were fixed in 10% neutral buffered formalin overnight, embedded in paraffin, and 7 µm sections were taken, and stained as we described (*Chang et al., 2007*; *Kim et al., 2013*). Sections were deparaffinized and antigen retrieval was performed as described previously (*Chang et al., 2007*; *Kim et al., 2013*). Sections were then permeabilized using 0.1% Triton X100 in PBS for 15 min, blocked for 2 hr with 10% goat serum, and incubated with anti-KLK10 (BiossUSA bs-2531R, 1:100) or anti-CD31 (Abcam ab28364, 1:100) primary antibody overnight at 4°C followed by Alexa Fluor-647 (Thermo Fisher Scientific, 1:500) secondary antibody for 2 hr at room temperature. Nuclei were counterstained with VectaShield that contained DAPI (Vector Laboratories). All confocal images were taken with a Zeiss (Jena, Germany) LSM800 confocal microscope. Endothelial KLK10 fluorescent intensity was measured with NIH ImageJ using CD31 as a reference.

## Serum lipid analysis

Serum lipid analysis was performed at the Cardiovascular Specialty Laboratories (Atlanta, GA) using a Beckman CX7 biochemical analyzer for total cholesterol, triglycerides, HDL and LDL as we reported (*Son et al., 2013*).

## Statistical analyses

Statistical analyses were performed using GraphPad Prism software. All of the n numbers represent biological replicates. Error bars depict the standard error of means (SEMs). Initially, the datasets were analyzed for normality using the Shapiro–Wilk test ($p < 0.05$) and equal variance using the *F*-test ($p > 0.05$). Data that followed a normal distribution and possessed equal variance were analyzed using two-tailed Student *t*-test or one-way analysis of variance (ANOVA), where appropriate with Bonferroni post hoc test as needed. In the case where the data showed unequal variances, an unpaired *t*-test with Welch correction was performed or Brown–Forsythe and Welch ANOVA for multiple comparisons. In the case where the data failed the Shapiro–Wilk test ($p > 0.05$), a nonparametric Mann–Whitney *U*-test was conducted for pairwise comparisons or the Kruskal–Wallis for multiple groups was performed.

## Acknowledgements

This work was supported by funding from the National Institutes of Health grants HL119798 and HL139757 to HJ. DW was supported by the NIH F31 HL145974 grant. HJ was also supported by John and Jan Portman Professorship and Wallace H Coulter Distinguished Faculty Professorship. LOM was supported by the NIH grants HL104165 and HL142975. EWT and LZ thank Cancer Research UK for support (grant C24523/A25192). JP was supported by the Centers for Disease Control and Prevention (CDC) LaSSI 201706. Excellent technical support of Priyanka Shakamuri and Jaclyn Weinberg at the CDC in synthesizing PAR1 and PAR2 peptides is acknowledged.

## Additional information

### Competing interests

Hwakyoung Lee, Yongjin An: is affiliated with Celltrion. The author has no financial interests to declare. Eleftherios P Diamandis: has consulted for Abbott diagnostics and Imaware Disgnostics. The author has no other competing interest to declare. Hanjoong Jo: is the founder of FloKines Pharma. The other authors declare that no competing interests exist.

### Funding

| Funder | Grant reference number | Author |
|---|---|---|
| National Heart, Lung, and Blood Institute | HL119798 | Hanjoong Jo |
| National Heart, Lung, and Blood Institute | HL145974 | Darian Williams |
| National Heart, Lung, and Blood Institute | HL139757 | Hanjoong Jo |
| Wallace H Coulter Foundation | | Hanjoong Jo |
| National Institutes of Health | HL104165 and HL142975 | Laurent O Mosnier |
| Cancer Research UK | C24523/A25192 | Edward W Tate Leran Zhang |
| Centers for Disease Control and Prevention | LaSSI 201706 | Jan Pohl |

The funders had no role in study design, data collection, and interpretation, or the decision to submit the work for publication.

### Author contributions

Darian Williams, Conceptualization, Data curation, Formal analysis, Funding acquisition, Investigation, Methodology, Project administration, Supervision, Validation, Writing – original draft, Writing – review and editing; Marwa Mahmoud, Conceptualization, Data curation, Formal analysis, Investigation, Methodology, Project administration, Resources, Supervision, Validation, Writing – original draft,

Writing – review and editing; Renfa Liu, Conceptualization, Data curation, Formal analysis, Methodology, Resources, Validation; Aitor Andueza, Conceptualization, Data curation, Formal analysis, Methodology, Supervision, Validation, Writing – review and editing; Sandeep Kumar, Jiahui Zhang, Conceptualization, Data curation, Formal analysis, Methodology, Resources, Supervision, Validation, Writing – review and editing; Dong-Won Kang, Conceptualization, Data curation, Formal analysis, Investigation, Methodology, Validation, Writing – review and editing; Ian Tamargo, Data curation, Formal analysis, Investigation, Methodology, Resources; Nicolas Villa-Roel, Data curation, Formal analysis, Methodology, Resources, Validation; Kyung-In Baek, Hwakyoung Lee, Formal analysis, Methodology, Resources; Yongjin An, Formal analysis, Resources; Leran Zhang, Data curation, Formal analysis, Methodology, Resources; Edward W Tate, Eleftherios P Diamandis, Resources, Supervision; Pritha Bagchi, Jan Pohl, Conceptualization, Data curation, Formal analysis, Methodology, Resources, Writing – review and editing; Laurent O Mosnier, Resources; Koichiro Mihara, Resources, Supervision, Validation; Morley D Hollenberg, Methodology, Resources, Supervision, Validation; Zhifei Dai, Methodology, Resources, Supervision; Hanjoong Jo, Conceptualization, Funding acquisition, Investigation, Methodology, Project administration, Supervision, Writing – original draft, Writing – review and editing

### Author ORCIDs

Darian Williams http://orcid.org/0000-0002-4572-3056
Nicolas Villa-Roel http://orcid.org/0000-0002-2981-9330
Eleftherios P Diamandis http://orcid.org/0000-0002-1589-820X
Hanjoong Jo http://orcid.org/0000-0003-1833-372X

### Ethics

Human coronary arteries were obtained from deidentified human hearts not suitable for cardiac transplantation donated to LifeLink of Georgia. Therefore, Emory University determined that this study was an IRB-exempt study.

All animal studies were performed with male C57BL/6 or ApoE−/− mice (Jackson Laboratory), were approved by Institutional Animal Care and Use Committee by Emory University (PROTO201700428), and were performed in accordance with the established guidelines and regulations consistent with federal assurance.

### Decision letter and Author response

Decision letter https://doi.org/10.7554/eLife.72579.sa1
Author response https://doi.org/10.7554/eLife.72579.sa2

### Data availability

All data generated or analyzed during this study are included in the manuscript and supporting file; Source Data files for all western blots and gels have been provided for all applicable figures. Previously Published Datasets: Endothelial reprogramming by disturbed flow revealed by single-cell RNAseq and chromatin accessibility study: Andueza A, Kumar S, Kim J, Kang DW, Mumme HL, Perez JI, Villa-Roel N, Jo H, 2020, https://www.ncbi.nlm.nih.gov/bioproject/?term=PRJNA646233, NCBI Bioproject, PRJNA646233.

The following previously published datasets were used:

| Author(s) | Year | Dataset title | Dataset URL | Database and Identifier |
|---|---|---|---|---|
| Andueza A, Kumar S, Kim J, Kang DW, Mumme HL, Perez JI, Villa-Roel N, Jo H | 2020 | Endothelial reprogramming by disturbed flow revealed by single-cell RNAseq and chromatin accessibility study | https://www.ncbi.nlm.nih.gov/bioproject/?term=PRJNA646233 | NCBI BioProject, PRJNA646233 |

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
