## [Editor Report]

This group has previously demonstrated that endothelial expression of kallikrein related-peptidase 10 (KLK10) is elevated arteries in conditions of high stable flow and down-regulated by disturbed flow conditions. In the present study, the authors tested the anti-atherogenic effects of KLK10 and found that endothelial expression of KLK10 after artery ligation or exposure to oscillatory flow decreases. Notably, recombinant KLK10 or KLK10 overexpression reproduce many vaso-protective effects and reduced atherosclerosis in a mouse model. Besides new insights into the atherogenic process, a potential therapeutic target may have been discovered.

---

## [Decision Letter]

**Decision letter after peer review:**

Thank you for submitting your article "Stable Flow-induced Expression of KLK10 Inhibits Endothelial Inflammation and Atherosclerosis" for consideration by *eLife*. Your article has been reviewed by 3 peer reviewers, and the evaluation has been overseen by a Reviewing Editor and a Senior Editor. The reviewers have opted to remain anonymous.

The reviewers have discussed their reviews with one another, and the Reviewing Editor has drafted this letter to help you prepare a revised submission.

Essential revisions:

1) More clarity supporting the claimed endothelial specific expression of KLK10 with better images and clearer explanations of VCAM1 quantification.

2) Also, in regard to VCAM1, a monocyte adhesion assay to strengthen the VCAM1 expression data.

3) It is essential to provide additional data to support the role of KLK10 in the inhibition of endothelial inflammation (which is also related to the first 2 points).

4) The data on the roles of HTRA1 and PAR are considered incomplete. Reviewers suggested either additional studies to strengthen the case for them, or to remove the data (or include as supplemental data). If stronger data are not available, statements about their roles should be toned down and Figure 9 eliminated from the main text, with mention that definitive conclusions on their roles will require more data in future studies.

*Reviewer #1:*

This is an interesting study that combines in vitro and in vivo approaches to bridge the gap between mechanosensitive endothelial cells and atherosclerotic plaque formation. The work stems from previous discoveries from the authors' laboratory and proceeds to expand and refine those findings. The authors make interesting connections between the major protagonists, KLK10, PAR1/PAR2, and HTRA1. Yet they did not completely establish how these elements come together to regulate inflammation and plaque progression.

Specific points are noted below:

In figure 1b, it is surprising that RCA imaging shows KLK10 expression in the sub-endothelial compartment, but authors comment that KLK10 is expressed uniquely in endothelium.

Are the endothelial effects on adhesion molecule expression and permeability specific to KLK10 or do they extend to other family members (KLK2, KLK4)?

The effect of rKLK10 appears to occur within a very narrow concentration range. Does this correspond to physiological concentrations?

rKLK10 reduced atherosclerosis in the partially ligated LCA. Was this effect duplicated in the aortic arch of the same animals? Whole mount photographs of vessels (4a) are too dim to decipher if this is the case.

It would be important to confirm proximity ligation assay result by co-IP.

Do VEGFA or VATB2 siRNA also reduce anti-inflammatory effects of KLK10? Could either molecule account for the residual protective effect of KLK10 after HTRA1 siRNA?

Does HTRA1 influence PAR1/PAR2 cleavage? The link between KLK10 activity and PAR1/PAR2 remains unresolved.

The work done so far is compelling. Nevertheless, it would be interesting to verify how PAR1/2 cleavage occurs. Is it possible that HTRA1 itself cleaves PARs? This possibility has not been investigated.

*Reviewer #2:*

Williams et al., demonstrate that Klk10 is upregulated under laminar flow conditions and downregulated in disturbed flow conditions both in vitro and in vivo. They further show that Klk10 overexpression or recombinant Klk10 administration reduces monocyte adhesion to endothelial cells, lessens endothelial permeability, and reduces inflammatory induction of ICAM1 and VCAM1 -- all of which are canonical indicators of endothelial activation. The authors then demonstrate in a mouse model of atherosclerosis that exogenous Klk10 reduces plaque development in a cholesterol and triglyceride independent manner, which is consistent with the hypothesis that the anti-atherosclerotic effect of Klk10 is mediated in/on the endothelium.

The investigators go on to show that the ability of overexpressed and recombinant Klk10 to reduce monocyte adhesion depends on PAR1 and PAR2, implicating these receptors as targets of Klk10; they further demonstrate that while Klk10 has serine protease activity, overexpressed/recombinant Klk10 does not induce the cleavage of PAR1 or PAR2.

The authors then determine through affinity pulldown and mass spectrometry that HTRA1, VEGFA, and VATB2 bind to Klk10. They further investigate HTRA1 and find that knockdown of HTRA1 results in increased endothelial activation. Additionally, they find that HTRA1 is upregulated by oscillatory shear (OS) relative to linear shear (LS), which is opposite the effect OS and LS have on Klk10.

Lastly, Williams et al., find that Klk10 levels are higher in advanced human atherosclerotic plaques that in early plaques.

These results demonstrate that Klk10 is an import regulator of endothelial activation and atherosclerotic plaque development. The authors elucidate a mechanism by which HTRA1 may regulate the activity of Klk10.

Comments

The Figure 1 legend contains references that haven't been given numeric symbols. Also, clusters E1 and E4 are not explained in the text or the legend.

Figure 3c legend should contain the phrase "subjected to OS" rather than "subjected OS."

Discussion line 7 should read "Unexpectedly, however, KLK10 did" rather than "Unexpectedly, however, but KLK10 did."

Figure S9 legend contains red underlines in the figure labels.

Figure S13 legend contains a typo. "Firboblasts" should be "Fibroblasts."

The anti-inflammatory effect of Klk10 is proposed by the authors to be mediated by PAR1 and PAR2. However, the authors only assay monocyte adhesion, whereas in other sections of the paper, endothelial permeability and the expression of VCAM1 and ICAM1 are used as additional measures of endothelial activation. Do PAR1 and PAR2 not mediate the effect of Klk10 on these other phenotypes?

The effect of oscillatory shear on HTRA1 is demonstrated in vitro (Figure 7e, S12a) but not in vivo (Figure S13a). The expression of Klk10 and Htra1 appear to be largely concordant among the scRNA-seq endothelial clusters from Andueza et al., (except for E8). Additionally, endothelial activation appears to be increased by HTRA1 knockdown (Figure 7d-g) in OS conditions, which seemingly contradicts the results in Figure 7d-g. Lastly, the experiment in Figure 7h-i was performed under LS conditions, which is different than the conditions under which the other endothelial activation experiments were performed (Figure 7d-g).

*Reviewer #3:*

The authors showed previously that the peptidase KLK10 is expressed by endothelial cells in response to stable laminar flow whereas its expression is suppressed under disturbed flow. They now report on an exploration of the potential function of KLK10 in endothelial cells and provide some evidence that KLK10 is involved in mediating the anti-inflammatory and anti-atherogenic effect of stable laminar flow. The study is of potential interest; however, some of the conclusions need to be better supported by experimental data, the underlying mechanism remains still rather unclear, and there are also some conceptual issues.

1. A large part of the experiments is based on the use of recombinant KLK10. It is not clear whether the employed concentrations are similar to those found under physiological conditions. It may be difficult to measure them in tissues, but a rough estimation would be helpful to understand this a little better. In Figure 2a, it looks as if KLK10 expression has an effect on the basal adhesion of monocytes (in the absence of TNFα). This should be tested directly. The effect of recombinant KLK10 appears to be smaller (Figure 2b). Is there an explanation for this difference?

2. When giving recombinant KLK10 systemically (e.g. Figure 2j and k, Figure 4), the authors gave injections (where?) every two days or twice per week. What is the plasma half-life of KLK10? It is important to know how long KLK10 is present at particular concentrations in the blood to interpret the effects of the chosen dosage scheme.

3. Data presented in Figure 2j and k are not clear. How was VCAM1 expression quantified? It would be good to have a counterstain to normalize expression. In any case, the subtle effect of recombinant KLK10 on endothelial VCAM1 expression in the greater curvature shown in the statistical analysis (Figure 2k) is not reflected by the image (Figure 2j), which shows a dramatic reduction in VCAM1 staining. Please explain how staining intensity was compared between different groups of animals. Also the graph in Figure 2k is unclear. The figure legend says that VCAM1 expression is presented as fold change normalized to control LC condition, but the numbering of the ordinate shows a value of about 125.

4. For the experiments shown in Figure 5 it is very important to prove expression of KLK10 in endothelial cells of the carotid artery. The data presented in Figure 5f are not clear, and the images are of rather poor quality. Higher magnified images should be shown. Why is there a strong signal for KLK10 in the adventitia in the test group but not in the Luc control?

5. The experiments shown in Figure 6a and b are key experiments testing an involvement of PAR1/2 in the effects of KLK10. While the authors used both, a knockdown and a pharmacological inhibitor approach, they only tested effects on monocyte adhesion. This needs to be extended to other readouts including downstream signaling (NFκB) and expression of inflammatory genes to rule out unspecific effects on one readout system. Do the authors have an explanation why inhibition of PAR2 in combination with recombinant KLK10 leads to an increase in monocyte adhesion whereas both alone have no effect?

6. The data indicating a role of HTRA1 in the effect of KLK10 are for various reasons still not clear and certainly not strong enough to support the conclusion. The authors show that HTRA1 can cleave KLK10, but it remains completely unclear what the consequences of this are. Somehow the authors seem to propose that cleavage of KLK10 is required for KLK10 to activate PAR receptors (at least, that is what Figure 9 and the corresponding figure legend suggests). However, no experiments have been performed to support this. It should be possible to test whether cleaved KLK10 fragments function as activators of PARs.

7. I also have a general conceptual problem: If it is true that the effect of KLK10 depends on HTRA1, it is difficult to understand how this would function under in vivo conditions, since the effect of KLK10 is in particular relevant under conditions of laminar stable flow when KLK10 is upregulated, whereas under these conditions HTRA1 expression is low or hardly measureable (e.g. blot in Figure 7e). Since the relationship of HTRA1 and KLK10 is studied using exogenously added recombinant KLK10, direct or indirect functional interactions can be seen, but it remains completely unclear whether they occur also when endogenously expressed proteins are studied.

8. Vascular PAR receptors seem to have complex mechanisms and roles in the progression of atherosclerosis and some studies show pro- as well as antiatherogenic/antiinflammatory effects (e.g. Minami et al., Arterioscler. Thromb. Vasc. Biol. 24:41 (2004); Kim et al., Sci. Rep. 8:15172 (2018); Seitz et al., Arterioscler. Thromb. Vasc. Biol. 27:769 (2007); Archiniegas et al., DNA Cell Biol. 23:815 (2004)). The authors argue solely on the basis of an anti-inflammatory antiatherogenic role of endothelial PAR1/2. This point needs to be discussed adequately.

---

## [Author Response]

Essential revisions:1) More clarity supporting the claimed endothelial specific expression of KLK10 with better images and clearer explanations of VCAM1 quantification.

To address this important point about endothelial-specific KLK10 expression, we carried out a new co-immunostaining study using antibodies to CD31 (as an endothelial cell marker) and KLK10. As shown in Figures 1b and 6f, KLK10 protein is found in the luminal endothelial cells, but it was also found in the adventitia and occasionally observed in the subendothelial layer as well (Figures 1b and 6f). In addition, KLK10 is also found throughout the LCA plaques in mice overexpressing KLK10 plasmid (Figure 6f). It is important to note that our single-cell RNAseq and ATACseq analyses of KLK10 expression in the mouse carotid artery clearly demonstrate that KLK10 mRNA is highly expressed only in endothelial cells but not in other cell types including the smooth muscle cells, fibroblasts, and immune cells (Figures 1K and 1L; Figures S8a and S9a). In addition, KLK10 is a secreted protein, which could be released to the circulation to be found in other locations including the adventitia and diffuse to the sub-endothelial layer. Therefore, we conclude that KLK10 protein signals observed in the sub-endothelial and adventitial layers are most likely not to be originated from the residential (non-endothelial) cells, while KLK10 is directly produced in endothelial cells.

These changes have been made as follows:

Result for Figure 1

“Importantly, all non-endothelial cell types in the carotid artery express nearly undetectable levels of KLK10 mRNA transcript and also display closed chromatin accessibility in the KLK10 promoter region, demonstrating that KLK10 is primarily expressed by ECs. This suggests that KLK10 protein observed in non-endothelial layers, including the adventitia and sub-endothelial layer (Figure 1b), is unlikely to be originated from cell types other than ECs.”

Discussion

“The KLK10 mRNA transcript was primarily found in ECs, while KLK10 protein was found not only in ECs but also in the adventitia and subendothelial layer (Figure 1). It is important to note that our single-cell RNAseq and ATACseq analyses of KLK10 expression in the mouse carotid artery clearly demonstrate that KLK10 mRNA is highly expressed only in endothelial cells but not in other cell types including the smooth muscle cells, fibroblasts, or immune cells (Figures 1K and 1L; Figures S8a and S9a). In addition, KLK10 is a secreted protein, which could be released to the circulation to be found in other locations including the adventitia and diffuse to the subendothelial layer. Therefore, we conclude that KLK10 protein signals observed in the subendothelial and adventitial layers are likely to be originated from ECs.”

To address the comment regarding the clearer explanations of the VCAM1 quantification, we reanalyzed the staining result and replaced the images with representative images (Figures 2j and S5a). We used three Z-sections showing the endothelial layer using the internal elastic laminar as a reference from each tissue samples to quantify VCAM1 expression in the intimal layer (Orthogonal image shown in Figure S6). The VCAM1 fluorescence intensity was quantified using the NIH Image J program as detailed in the Methods. We found that rKLK10 treatment showed a decreasing trend in the VCAM1 expression in the greater curvature region, but it did not reach statistical significance (Figure 2k). Thank you for pointing out an error in the Figure 2k label, which we corrected in the new Figure 2k.

These changes have been made in Figures 2j, 2k, S5a, and S6 and addressed in the accompanying Methods section as follows:

“We used three Z-sections showing the endothelial layer using the internal elastic laminar as a reference from each tissue samples to quantify VCAM1 or KLK10 expression in the ECs (Orthogonal image shown in Figure S6). The fluorescence intensity was quantified using the NIH Image J program.”

2) Also, in regard to VCAM1, a monocyte adhesion assay to strengthen the VCAM1 expression data.

We would like to clarify that we previously showed the effect of rKLK10 and plasmid-derived KLK10 overexpression on monocyte adhesion and VCAM1 expression in HAECs (Figure 2a-i). In response to this comment and Reviewer #1- C2, we carried out an additional study to compare the anti-inflammatory effect of rKLK10 to rKLK8 and rKLK11. We chose these two others since they are the only other KLK family members highly expressed in endothelial cells (Figure S10) and are also closely related to KLK10^1^. We found that rKLK10, but not rKLK8 and rKLK11, inhibited monocyte adhesion and VCAM1 expression in response to TNFa in HAECs (Figure S12). These results suggest the unique anti-inflammatory effect of KLK10.

3) It is essential to provide additional data to support the role of KLK10 in the inhibition of endothelial inflammation (which is also related to the first 2 points).

Since our original manuscript contained so much data, some of the results presented as Supplementary data may not have been obvious to the reviewers. To address this comment and also in response to the Editor’s recommendation to remove PAR1/2 and HTRA1 data from the manuscript, we have moved the old Supplement Figure demonstrating that the role of p65 NFkB signaling pathway in the anti-inflammatory effect of KLK10 to the new Figure 3 in the revised manuscript. We found that KLK10 inhibits the shear- and TNFa-induced NFkB pathway by preventing p65 phosphorylation and p65 nuclear localization (Figure 3).

This data has been added in the Results section as Figure 3 and is described further in the accompanying methods section.

4) The data on the roles of HTRA1 and PAR are considered incomplete. Reviewers suggested either additional studies to strengthen the case for them, or to remove the data (or include as supplemental data). If stronger data are not available, statements about their roles should be toned down and Figure 9 eliminated from the main text, with mention that definitive conclusions on their roles will require more data in future studies.

Since our original manuscript contained so much data, some of the results presented as Supplementary data may not have been obvious to the reviewers. To address this comment and also in response to the Editor’s recommendation to remove PAR1/2 and HTRA1 data from the manuscript, we have moved the old Supplement Figure demonstrating that the role of p65 NFkB signaling pathway in the anti-inflammatory effect of KLK10 to the new Figure 3 in the revised manuscript. We found that KLK10 inhibits the shear- and TNFa-induced NFkB pathway by preventing p65 phosphorylation and p65 nuclear localization (Figure 3).

This data has been added in the Results section as Figure 3 and is described further in the accompanying methods section.

Reviewer #1:This is an interesting study that combines in vitro and in vivo approaches to bridge the gap between mechanosensitive endothelial cells and atherosclerotic plaque formation. The work stems from previous discoveries from the authors' laboratory and proceeds to expand and refine those findings. The authors make interesting connections between the major protagonists, KLK10, PAR1/PAR2, and HTRA1. Yet they did not completely establish how these elements come together to regulate inflammation and plaque progression.Specific points are noted below:In figure 1b, it is surprising that RCA imaging shows KLK10 expression in the sub-endothelial compartment, but authors comment that KLK10 is expressed uniquely in endothelium.

Thank you for your insightful and constructive comments. To address this important point about endothelial-specific KLK10 expression, we carried out a new co-immunostaining study using antibodies to CD31 (as an endothelial cell marker) and KLK10. As shown in Figures 1b and 6f, KLK10 expression is found in the luminal endothelial cells and in the adventitia. Also, KLK10 is occasionally observed in the subendothelial layer as well (Figures 1b and 6f). In addition, KLK10 is also found throughout the LCA plaques in mice overexpressing KLK10 plasmid (Figure 6f). It is important to note that our single-cell RNAseq and ATACseq analyses of KLK10 expression in the mouse carotid artery clearly demonstrate that KLK10 mRNA is highly expressed only in endothelial cells but not in other cell types including the smooth muscle cells, fibroblasts, and immune cells (Figures 1K and 1L; Figures S8a and S9a). In addition, KLK10 is a secreted protein, which could be released to the circulation to be found in other locations including the adventitia and diffuse to the sub-endothelial layer. Therefore, we conclude that KLK10 protein signals observed in the sub-endothelial and adventitial layers are most likely not to be originated from the residential cells, while KLK10 is directly produced in endothelial cells.

The new co-immunostaining study has been added as Figure 6f and addressed in the discussion and Results section as follows:

Results:

“Importantly, all non-endothelial cell types in the carotid artery express nearly undetectable levels of KLK10 mRNA transcript and also display closed chromatin accessibility in the KLK10 promoter region, demonstrating that KLK10 is primarily expressed by ECs. This suggests that KLK10 protein observed in non-endothelial layers, including the adventitia and sub-endothelial layer (Figure 1b), is unlikely to be originated from cell types other than ECs.”

Discussion:

“The KLK10 mRNA transcript was primarily found in ECs, while KLK10 protein was found not only in ECs but also in the adventitia and subendothelial layer (Figure 1). It is important to note that our single-cell RNAseq and ATACseq analyses of KLK10 expression in the mouse carotid artery clearly demonstrate that KLK10 mRNA is highly expressed only in endothelial cells but not in other cell types including the smooth muscle cells, fibroblasts, or immune cells (Figures 1K and 1L; Figures S8a and S9a). In addition, KLK10 is a secreted protein, which could be released to the circulation to be found in other locations including the adventitia and diffuse to the subendothelial layer. Therefore, we conclude that KLK10 protein signals observed in the subendothelial and adventitial layers are likely to be originated from ECs.”

Are the endothelial effects on adhesion molecule expression and permeability specific to KLK10 or do they extend to other family members (KLK2, KLK4)?

In response to this comment, we carried out an additional study to compare the anti-inflammatory effect of rKLK10 to rKLK8 and rKLK11. We chose these two others since they are the only other KLK family members highly expressed in endothelial cells (Figure S10) and are also closely related to KLK10^1^. We found that rKLK10, but not rKLK8 or rKLK11, inhibited monocyte adhesion and VCAM1 expression in response to TNFa in HAECs (Figure S12).

These results suggest the unique anti-inflammatory effect of KLK10.

These changes have been made as Figure S12 and are discussed in the Discussion as follows:

“Interestingly, the anti-inflammatory effect of KLK10 seem to be unique in comparison to other KLKs expressed in ECs, including KLK8 and KLK11. Analysis of the sc-RNAseq dataset showed that KLK8 and KLK11 are two other KLK members expressed in ECs (Figure S12). We found that rKLK10, but not rKLK8 or rKLK11, inhibited endothelial inflammation in response to TNFa in HAECs (Figure S12).”

The effect of rKLK10 appears to occur within a very narrow concentration range. Does this correspond to physiological concentrations?

Overall, the effective concentration of rKLK10 we used in this study is within a reasonable range of human and mouse KLK10 levels in the plasma. Our mouse KLK10 ELISA study (Figure S7) showed that KLK10 in ApoE^-/-^ mouse plasma is in the range 5-10 ng/mL. In humans, normal mean plasma KLK10 levels to be ~0.5 ng/mL, with a range from nearly undetectable to ~20 ng/mL in various cancers patients^2, 3^. We found that KLK10 level in HAECs exposed to the anti-inflammatory laminar shear stress (LS) was ~0.3 ng/mL, which was decreased to ~0.13 ng/mL by the pro-inflammatory oscillatory shear (OS) (Figure 1j).

We found that 1-10 ng/mL of rKLK10 inhibits permeability and inflammation in HAECs, which falls within the reasonable physiological range. The effective rKLK10 dose used in mouse studies was 0.6 mg/Kg. We carried out a new study to determine the plasma KLK10 levels following the injection of KLK10 at 0.6 mg/Kg dose, which produced a peak (~1,600 ng/mL) with a t_1/2_ of 4.5 hr, becoming undetectable by 24 hr (Figure S5c). This transient level of human rKLK10 in the mouse study makes it difficult to correlate it to a murine physiological levels.

These changes have been made as Figure S5c and addressed in the Discussion as follows:

“KLK10 expression is downregulated in breast, prostate, testicular, and lung cancer^25-29^ but overexpressed in ovarian, pancreatic, and uterine cancer^30-33^. These suggest that abnormal, either too low or too high, levels of KLK10 are associated with various pathophysiological conditions. Overall, the effective concentration of rKLK10 we used in this study is within a reasonable range of human and mouse KLK10 levels in the plasma. Our mouse KLK10 ELISA study (Figure S7) showed that plasma KLK10 level in ApoE^-/-^ mice is in the range of 5-10 ng/mL. In humans, normal plasma KLK10 levels are ~0.5 ng/mL, with a range from nearly undetectable to ~20 ng/mL in various cancers patients^30, 53^. We found that KLK10 levels in HAECs exposed to the anti-inflammatory LS was ~0.3 ng/mL, which decreased to ~0.13 ng/mL by the proinflammatory OS (Figure 1j). In functional studies, we found that 1-10 ng/mL of rKLK10 inhibits permeability and inflammation in HAECs, which falls within the reasonable physiological range. The effective rKLK10 dose used in mouse studies was 0.6 mg/Kg, although how this effective dose translates to humans will need to be further studied.”

rKLK10 reduced atherosclerosis in the partially ligated LCA. Was this effect duplicated in the aortic arch of the same animals? Whole mount photographs of vessels (4a) are too dim to decipher if this is the case.

The design of the study does not allow for a proper assessment of plaque in the aortic arch, as this was an acute study performed at a 3-week timepoint. A chronic study with ApoE^-/-^ on high-fat diet for 3 months would be necessary to study the effect of rKLK10 in the aortic arch. Given the potential complications of using the repeated rKLK10 iv injections over an extended time, we would need a different delivery approach to determine its effect on atherosclerosis in the aortic arch. While this is our future plan, this study is beyond the scope of this study. To address the dimness of the images, we present improved images in the new Figure 5a.

These changes have been made as Figure 5a.

It would be important to confirm proximity ligation assay result by co-IP.

As suggested by the Editor in C#4, we have removed all data related to the KLK10 binding proteins including HTRA1 and revised the Results and Discussion accordingly.

Do VEGFA or VATB2 siRNA also reduce anti-inflammatory effects of KLK10? Could either molecule account for the residual protective effect of KLK10 after HTRA1 siRNA?

As suggested by the Editor in C#4, we have removed all data related to KLK10 binding proteins including HTRA1, VEGFA, and VATB2 and revised the Results and Discussion accordingly.

Does HTRA1 influence PAR1/PAR2 cleavage? The link between KLK10 activity and PAR1/PAR2 remains unresolved.

Yes, it does, but as suggested by the Editor, we have removed all data related to HTRA1 and PAR1/2.

Reviewer #2:CommentsThe Figure 1 legend contains references that haven't been given numeric symbols. Also, clusters E1 and E4 are not explained in the text or the legend.

We deeply appreciate your thorough and constructive comments. This has been corrected in the resubmission as marked in Red. Additional description for EC clusters have now been added (new Figure 1k and 1l; Figures S8-S10).

Figure 3c legend should contain the phrase "subjected to OS" rather than "subjected OS."

This has been corrected in the resubmission.

Discussion line 7 should read "Unexpectedly, however, KLK10 did" rather than "Unexpectedly, however, but KLK10 did."

This has been corrected in the resubmission.

Figure S9 legend contains red underlines in the figure labels.

This has been corrected in the resubmission.

Figure S13 legend contains a typo. "Firboblasts" should be "Fibroblasts."

This has been corrected in the resubmission.

The anti-inflammatory effect of Klk10 is proposed by the authors to be mediated by PAR1 and PAR2. However, the authors only assay monocyte adhesion, whereas in other sections of the paper, endothelial permeability and the expression of VCAM1 and ICAM1 are used as additional measures of endothelial activation. Do PAR1 and PAR2 not mediate the effect of Klk10 on these other phenotypes?

As suggested by the Editor in C#4, we have removed all data related to PAR1/2 and revised the Results and Discussion accordingly.

The effect of oscillatory shear on HTRA1 is demonstrated in vitro (Figure 7e, S12a) but not in vivo (Figure S13a). The expression of Klk10 and Htra1 appear to be largely concordant among the scRNA-seq endothelial clusters from Andueza et al., (except for E8). Additionally, endothelial activation appears to be increased by HTRA1 knockdown (Figure 7d-g) in OS conditions, which seemingly contradicts the results in Figure 7d-g.

As suggested by the Editor, we have removed all data related to HTRA1 and revised the Results and Discussion accordingly.

Lastly, the experiment in Figure 7h-i was performed under LS conditions, which is different than the conditions under which the other endothelial activation experiments were performed (Figure 7d-g).

Thank you for pointing out the different shear conditions used to study p65 NFkB nuclear translocation. We should have provided the rationale why we used high laminar shear (LS) to activate NFkB pathway in endothelial cells in our study. Numerous studies have shown that NFkB activation, including p65 translocation to the nucleus, is induced by either LS or OS in endothelial cells^4-12^. However, most shear studies use LS conditions to induce p65 translocation in a rapid and robust manner compared to the OS, which requires prolonged shear exposure. Therefore, we also used LS conditions to study robust NFkB activation including p65 translocation in an acute study.

This point has been addressed in Figure 3 and the accompanying Results section as follows:

“Since NFkB is a well-known pro-inflammatory transcription factor, which induces expression of VCAM1 and ICAM1 and subsequent monocyte adhesion to ECs^39-48^, we tested whether KLK10 inhibits NFkB activation in response to shear stress and TNFa. We first found that KLK10 prevented phosphorylation (p-Ser536) and trans-nuclear location of p65, two important markers of NFkB activation, in response to TNFa (Figure 3a-d). KLK10 also prevented trans-nuclear location of p65 in response to acute shear challenge using LS condition (Figure 3e,f), which is well-known to induce robust and transient NFkB activation^39-48^.”

Reviewer #3:The authors showed previously that the peptidase KLK10 is expressed by endothelial cells in response to stable laminar flow whereas its expression is suppressed under disturbed flow. They now report on an exploration of the potential function of KLK10 in endothelial cells and provide some evidence that KLK10 is involved in mediating the anti-inflammatory and anti-atherogenic effect of stable laminar flow. The study is of potential interest; however, some of the conclusions need to be better supported by experimental data, the underlying mechanism remains still rather unclear, and there are also some conceptual issues.1. A large part of the experiments is based on the use of recombinant KLK10. It is not clear whether the employed concentrations are similar to those found under physiological conditions. It may be difficult to measure them in tissues, but a rough estimation would be helpful to understand this a little better.

We sincerely appreciate your constructive and insightful comments. In the revision, we have addressed all of your comments.

This exact point was also raised by Reviewer #1 – Comment #3 and addressed above. Overall, the effective concentration of rKLK10 we used in this study is within a reasonable range of human and mouse KLK10 levels in the plasma. Our mouse KLK10 ELISA study (Figure S7) showed that KLK10 in ApoE^-/-^ mouse plasma is in the range 5-10 ng/mL. In humans, normal mean plasma KLK10 levels to be ~0.5 ng/mL, with a range from nearly undetectable to ~20 ng/ml in various cancers patients^2, 3^. We found that KLK10 level in HAECs exposed to the nti-inflammatory laminar shear stress (LS) was ~0.3 ng/mL, which was decreased to ~0.13 ng/mL by the pro-inflammatory oscillatory shear (OS) (Figure 1j).

We found that 1-10 ng/mL of rKLK10 inhibits permeability and inflammation in HAECs, which falls within the reasonable physiological range. The effective rKLK10 dose used in mouse studies was 0.6 mg/Kg, which produced a peak (~1,600 ng/mL) with a t_1/2_ of 4.5 hr, becoming undetectable by 24 hr (Figure S5c). This transient level of human rKLK10 in the mouse study makes it difficult to correlate it to a murine physiological levels.

These changes have been made as Figure S5c and addressed in the Discussion as follows:

“KLK10 expression is downregulated in breast, prostate, testicular, and lung cancer^25-29^ but overexpressed in ovarian, pancreatic, and uterine cancer^30-33^. These suggest that abnormal, either too low or too high, levels of KLK10 are associated with various pathophysiological conditions. Overall, the effective concentration of rKLK10 we used in this study is within a reasonable range of human and mouse KLK10 levels in the plasma. Our mouse KLK10 ELISA study (Figure S7) showed that plasma KLK10 level in ApoE^-/-^ mice is in the range of 5-10 ng/mL. In humans, normal plasma KLK10 levels are ~0.5 ng/mL, with a range from nearly undetectable to ~20 ng/mL in various cancers patients^30, 53^. We found that KLK10 levels in HAECs exposed to the anti-inflammatory LS was ~0.3 ng/mL, which decreased to ~0.13 ng/mL by the proinflammatory OS (Figure 1j). In functional studies, we found that 1-10 ng/mL of rKLK10 inhibits permeability and inflammation in HAECs, which falls within the reasonable physiological range. The effective rKLK10 dose used in mouse studies was 0.6 mg/Kg, although how this effective dose translates to humans will need to be further studied.”

In Figure 2a, it looks as if KLK10 expression has an effect on the basal adhesion of monocytes (in the absence of TNFα). This should be tested directly. The effect of recombinant KLK10 appears to be smaller (Figure 2b). Is there an explanation for this difference?

Because we presented so much data, the reviewer may have misunderstood the result in Figure 2a and overlooked the data Figure S2a. In Figure 2a, we tested the effect of KLK10 overexpression by plasmid vector on monocyte adhesion in the presence of TNFα. We also tested the effect of KLK10 on basal adhesion of monocytes (in the absence of TNFα), which was presented in Figure S2. This demonstrates that KLK10 inhibits monocyte adhesion in basal (Figure S2a) as well as TNFa-induced (Figure 2a) conditions.

Yes, it is true that the effect of rKLK10 on monocyte adhesion (Figure 2b) is weaker than that of KLK10 overexpression using the plasmid vector (Figure 2a). The exact mechanism is unclear. We speculate that KLK10 produced from plasmid directly in HAECs is processed to be more effective than the rKLK10 produced and processed in CHO cells, which underwent multiple purification steps and storage conditions.

2. When giving recombinant KLK10 systemically (e.g. Figure 2j and k, Figure 4), the authors gave injections (where?) every two days or twice per week. What is the plasma half-life of KLK10? It is important to know how long KLK10 is present at particular concentrations in the blood to interpret the effects of the chosen dosage scheme.

We apologize for the lack of clarity and inaccuracy. rKLK10 was administered by tail-vein every two days for five days for the study in Figure 2j and k. For atherosclerosis study in Figure 5, rKLK10 was injected via tail-vein once every three days for three weeks.

To address the important point on the KLK10 half-life in mouse plasma, we carried out a new study to determine plasma KLK10 level. For this study, we injected rKLK10 at 0.6 mg/Kg dose, which produced a peak (~1,600 ng/ml) with a t_1/2_ of 4.5 hr, reaching a low level by ~12 hr and becoming undetectable by 24 hr (Figure S5c). Therefore, injecting rKLK10 once every two or three days seems to be a reasonable dosage scheme to provide sufficient KLK10 levels in the plasma for the duration of the study.

Tail-vein injections have been clarified in the Results, Discussion, and Methods, as highlighted in red. The new half-life study has been added as Figure S5c and discussed in the results as follows:

**“**Injection of rKLK10 at 0.6 mg/Kg dose increased its plasma level to a peak of ~1,600 ng/ml with a t_1/2_ of 4.5 hr, becoming undetectable by 24 hr (Figure S5c).”

3. Data presented in Figure 2j and k are not clear. How was VCAM1 expression quantified? It would be good to have a counterstain to normalize expression. In any case, the subtle effect of recombinant KLK10 on endothelial VCAM1 expression in the greater curvature shown in the statistical analysis (Figure 2k) is not reflected by the image (Figure 2j), which shows a dramatic reduction in VCAM1 staining. Please explain how staining intensity was compared between different groups of animals. Also the graph in Figure 2k is unclear. The figure legend says that VCAM1 expression is presented as fold change normalized to control LC condition, but the numbering of the ordinate shows a value of about 125.

To address the comment regarding the clearer explanations of the VCAM1 quantification, we reanalyzed the staining result and replaced the images with representative images (Figures 2j and S5a). We used three Z-sections showing the endothelial layer using the internal elastic laminar as a reference from each tissue samples to quantify VCAM1 expression in the intimal layer (Orthogonal image shown in Figure S6). The VCAM1 fluorescence intensity was quantified using the NIH Image J program as detailed in the Methods. We found that rKLK10 treatment showed a decreasing trend in the VCAM1 expression in the greater curvature region, but it did not reach statistical significance (Figure 2k). Thank you for pointing out an error in the Figure 2k label, which we corrected in the new Figure 2k.

These changes have been made in Figures 2j, 2k, S5a, and S6 and addressed in the accompanying Methods section as follows:

“We used three Z-sections showing the endothelial layer using the internal elastic laminar as a reference from each tissue samples to quantify VCAM1 or KLK10 expression in the ECs (Orthogonal image shown in Figure S6). The fluorescence intensity was quantified using the NIH Image J program.”

4. For the experiments shown in Figure 5 it is very important to prove expression of KLK10 in endothelial cells of the carotid artery. The data presented in Figure 5f are not clear, and the images are of rather poor quality. Higher magnified images should be shown. Why is there a strong signal for KLK10 in the adventitia in the test group but not in the Luc control?

To address this important point about endothelial-specific KLK10 expression, we carried out a new co-immunostaining study using antibodies to CD31 (as an endothelial cell marker) and KLK10. As shown in Figures 1b and new Figure 6f (originally Figure 5f), KLK10 expression is found in the luminal endothelial cells and in the adventitia. Also, KLK10 is occasionally observed in the subendothelial layer as well (Figures 1b and 6f). In addition, KLK10 is also found throughout the LCA plaques in mice overexpressing KLK10 plasmid (Figure 6f). It is important to note that our single-cell RNAseq and ATACseq analyses of KLK10 expression in the mouse carotid artery clearly demonstrate that KLK10 mRNA is highly expressed only in endothelial cells but not in other cell types including the smooth muscle cells, fibroblasts, and immune cells (Figures 1K and 1L; Figures S8a and S9a). In addition, KLK10 is a secreted protein, which could be released to the circulation to be found in other locations including the adventitia and diffuse to the sub-endothelial layer. Therefore, we conclude that KLK10 protein signals observed in the sub-endothelial and adventitial layers are most likely not to be originated from the residential cells, while KLK10 is directly produced in endothelial cells.

The new co-immunostaining study has been added as Figure 6f and addressed in the discussion and Results section as follows:

Result for Figure 1

“Importantly, all non-endothelial cell types in the carotid artery express nearly undetectable levels of KLK10 mRNA transcript and also display closed chromatin accessibility in the KLK10 promoter region, demonstrating that KLK10 is primarily expressed by ECs. This suggests that KLK10 protein observed in non-endothelial layers, including the adventitia and sub-endothelial layer (Figure 1b), is unlikely to be originated from cell types other than ECs.”

Discussion

“The KLK10 mRNA transcript was primarily found in ECs, while KLK10 protein was found not only in ECs but also in the adventitia and subendothelial layer (Figure 1). It is important to note that our single-cell RNAseq and ATACseq analyses of KLK10 expression in the mouse carotid artery clearly demonstrate that KLK10 mRNA is highly expressed only in endothelial cells but not in other cell types including the smooth muscle cells, fibroblasts, or immune cells (Figures 1K and 1L; Figures S8a and S9a). In addition, KLK10 is a secreted protein, which could be released to the circulation to be found in other locations including the adventitia and diffuse to the subendothelial layer. Therefore, we conclude that KLK10 protein signals observed in the subendothelial and adventitial layers are likely to be originated from ECs.”

5. The experiments shown in Figure 6a and b are key experiments testing an involvement of PAR1/2 in the effects of KLK10. While the authors used both, a knockdown and a pharmacological inhibitor approach, they only tested effects on monocyte adhesion. This needs to be extended to other readouts including downstream signaling (NFκB) and expression of inflammatory genes to rule out unspecific effects on one readout system. Do the authors have an explanation why inhibition of PAR2 in combination with recombinant KLK10 leads to an increase in monocyte adhesion whereas both alone have no effect?

As suggested by the Editor’s Comment #4, we have removed all data related to PAR1/2 and revised the Results and Discussion accordingly.

6. The data indicating a role of HTRA1 in the effect of KLK10 are for various reasons still not clear and certainly not strong enough to support the conclusion. The authors show that HTRA1 can cleave KLK10, but it remains completely unclear what the consequences of this are. Somehow the authors seem to propose that cleavage of KLK10 is required for KLK10 to activate PAR receptors (at least, that is what Figure 9 and the corresponding figure legend suggests). However, no experiments have been performed to support this. It should be possible to test whether cleaved KLK10 fragments function as activators of PARs.

As suggested by the Editor’s Comment #4, we have removed all data related to PAR1/2 and HTRA1, and revised the Results and Discussion accordingly.

7. I also have a general conceptual problem: If it is true that the effect of KLK10 depends on HTRA1, it is difficult to understand how this would function under in vivo conditions, since the effect of KLK10 is in particular relevant under conditions of laminar stable flow when KLK10 is upregulated, whereas under these conditions HTRA1 expression is low or hardly measureable (e.g. blot in Figure 7e). Since the relationship of HTRA1 and KLK10 is studied using exogenously added recombinant KLK10, direct or indirect functional interactions can be seen, but it remains completely unclear whether they occur also when endogenously expressed proteins are studied.

As suggested by the Editor’s Comment #4, we have removed all data related to PAR1/2 and HTRA1, and revised the Results and Discussion accordingly.

8. Vascular PAR receptors seem to have complex mechanisms and roles in the progression of atherosclerosis and some studies show pro- as well as antiatherogenic/antiinflammatory effects (e.g. Minami et al., Arterioscler. Thromb. Vasc. Biol. 24:41 (2004); Kim et al., Sci. Rep. 8:15172 (2018); Seitz et al., Arterioscler. Thromb. Vasc. Biol. 27:769 (2007); Archiniegas et al., DNA Cell Biol. 23:815 (2004)). The authors argue solely on the basis of an anti-inflammatory antiatherogenic role of endothelial PAR1/2. This point needs to be discussed adequately.

As suggested by the Editor’s Comment #4, we have removed all data related to PAR1/2, and revised the Results and Discussion accordingly.References:

1. Stefanini ACB, da Cunha BR, Henrique T and Tajara EH. Involvement of KallikreinRelated Peptidases in Normal and Pathologic Processes. Dis Markers. 2015;2015:946572.

2. Luo L-Y, Katsaros D, Scorilas A, Fracchioli S, Bellino R, van Gramberen M, de Bruijn H, Henrik A, Stenman U-H, Massobrio M, van der Zee AGJ, Vergote I and Diamandis EP. The Serum Concentration of Human Kallikrein 10 Represents a Novel Biomarker for Ovarian Cancer Diagnosis and Prognosis. Cancer Res. 2003;63:807-811.

3. Planque C, Li L, Zheng Y, Soosaipillai A, Reckamp K, Chia D, Diamandis EP and Goodglick L. A multiparametric serum kallikrein panel for diagnosis of non–small cell lung carcinoma. Clin Cancer Res. 2008;14:1355-1362.

4. Baeriswyl DC, Prionisti I, Peach T, Tsolkas G, Chooi KY, Vardakis J, Morel S,

Diagbouga MR, Bijlenga P and Cuhlmann S. Disturbed flow induces a sustained, stochastic NFκB activation which may support intracranial aneurysm growth in vivo. Sci Rep. 2019;9:1-14. 5.

5. Baeyens N, Mulligan-Kehoe MJ, Corti F, Simon DD, Ross TD, Rhodes JM, Wang TZ, Mejean CO, Simons M and Humphrey J. Syndecan 4 is required for endothelial alignment in flow and atheroprotective signaling. Proceedings of the National Academy of Sciences. 2014;111:17308-17313.

6. Chen H, Wu L, Liu X, Chen Y and Wang B. Effects of laminar shear stress on IL-8 mRNA expression in endothelial cells. Biorheology. 2003;40:53-58.

7. Coleman PR, Lay AJ, Ting KK, Zhao Y, Li J, Jarrah S, Vadas MA and Gamble JR. YAP and the RhoC regulator ARHGAP18, are required to mediate flow-dependent endothelial cell alignment. Cell Communication and Signaling. 2020;18:1-12.

8. Lay AJ, Coleman PR, Formaz-Preston A, Ting KK, Roediger B, Weninger W, Schwartz MA, Vadas MA and Gamble JR. ARHGAP18: a flow-responsive gene that regulates endothelial cell alignment and protects against atherosclerosis. Journal of the American Heart Association. 2019;8:e010057.

9. Mohan S, Mohan N and Sprague EA. Differential activation of NF-kappa B in human aortic endothelial cells conditioned to specific flow environments. American Journal of Physiology-Cell Physiology. 1997;273:C572-C578.

10. Petzold T, Orr AW, Hahn C, Jhaveri KA, Parsons JT and Schwartz MA. Focal adhesion kinase modulates activation of NF-κB by flow in endothelial cells. American Journal of Physiology-Cell Physiology. 2009;297:C814-C822.

11. Wang Y, Flores L, Lu S, Miao H, Li Y-S and Chien S. Shear stress regulates the Flk1/Cbl/PI3K/NF-κB pathway via actin and tyrosine kinases. Cell Mol Bioeng. 2009;2:341-350.

12. Wilson AA, Kwok LW, Porter EL, Payne JG, McElroy GS, Ohle SJ, Greenhill SR, Blahna MT, Yamamoto K, Jean JC, Mizgerd JP and Kotton DN. Lentiviral delivery of RNAi for in vivo lineage-specific modulation of gene expression in mouse lung macrophages. Mol Ther. 2013;21:825-33.